**A Machine Learning Methodology for the Generation of a Parameterization of the Hydroxyl Radical**

Daniel C. Anderson[1,2], Melanie B. Follette-Cook[2,3], Sarah A. Strode[2,3], Julie M. Nicely[2,4], Junhua Liu[2,3], Peter D. Ivatt[2,4], Bryan N. Duncan[2]

[1] GESTAR II, University of Maryland Baltimore County, Baltimore, MD, USA
[2] Atmospheric Chemistry and Dynamics Laboratory, NASA Goddard Space Flight Center, Greenbelt, MD, USA
[3] GESTAR II, Morgan State University, Baltimore, MD, USA
[4] Earth System Science Interdisciplinary Center, University of Maryland, College Park, MD, USA

*Correspondence to*: Daniel C. Anderson (daniel.c.anderson@nasa.gov)

**Abstract**

We present a methodology that uses gradient boosted regression trees (a machine learning technique) and a full-chemistry simulation (i.e., training dataset) from a chemistry climate model (CCM) to efficiently generate a parameterization of tropospheric hydroxyl radical (OH) that is a function of chemical, dynamical, and solar irradiance variables. This surrogate model of OH is designed to be integrated into a CCM and allow for computationally-efficient simulation of nonlinear feedbacks between OH and tropospheric constituents that have loss by reaction with OH as their primary sinks (e.g., carbon monoxide (CO), methane ($CH_4$), volatile organic compounds (VOCs)). Such a model framework is advantageous for studies that require multi-decadal simulations of $CH_4$ or multi-year sensitivity simulations to understand the causes of trends and variations of CO and $CH_4$. To allow the user to easily target the training dataset towards a desired application, we are outlining a methodology to generate a parameterization of OH and not presenting an "off the shelf" version of a parameterization to be incorporated into a CCM. This provides for the relatively easy creation of a new parameterization in response to, for example, changes in research goals or the underlying CCM chemistry and/or dynamics schemes. We show that a sample parameterization of OH generated from a CCM simulation is able to reproduce OH concentrations with a normalized root mean square error of approximately 5%, as well as capturing the global mean methane lifetime within approximately 1%. Our calculated accuracy of the parameterization assumes inputs being within the bounds of the training dataset. Large excursions from these bounds will likely decrease the overall accuracy. However, we show that the sample parameterization predicts large deviations in OH for an El Niño event that was not part of the training dataset, and that the spatial distribution and strength of these deviations are consistent with the event. This result gives confidence in the fidelity of a parameterization developed with our methodology to simulate the spatial and temporal responses of OH to perturbations from large variations in the chemical, dynamical, and solar irradiance drivers of OH. In addition, we discuss how two machine learning metrics, Gain feature importance and SHAP values, indicate that the behavior of a parameterization of OH generally accords with our understanding of OH chemistry, even though there are no physics- or chemistry-based constraints on the parameterization.

**1.0 Introduction**

The hydroxyl radical (OH) is the dominant tropospheric oxidant. It removes numerous species from the atmosphere, including carbon monoxide (CO) and methane ($CH_4$), which are the largest OH sinks on a global scale (Spivakovsky et al., 2000;Spivakovsky et al., 1990). Recent trends in $CH_4$, the second most important anthropogenic greenhouse gas, can potentially be explained by changes in OH abundance (Rigby et al., 2017), although changes in emissions are also a likely contributor (Turner et al., 2017).

Likewise, the large increase in $CH_4$ during 2020 has been attributed to decreases in OH resulting from
COVID-19 related changes in $NO_X$ ($NO_X$ = NO + $NO_2$) abundance (Laughner et al., 2021).  Understanding
the non-linear chemistry of the drivers of OH and feedbacks among these species is therefore important
for characterizing past and present changes in the atmosphere as well as in projecting future climate
scenarios.
Chemistry-climate models (CCMs) with detailed chemical mechanisms are used to investigate the
complex, non-linear chemistry between these species and their impacts on the atmosphere (e.g.,Fiore et
al., 2006;Voulgarakis et al., 2015;Gaubert et al., 2017;Holmes, 2018).  The utility of CCMs for this
purpose is limited, however, by the large computational expense of a CCM with a full representation of
$O_3 - NO_X - VOC$ (Ozone, $NO_X$, Volatile Organic Compound) chemistry combined with the need to model
over decadal time scales in order to let $CH_4$ perturbations fully evolve (Prather, 1996).  Because of this
computational expense, simulations are necessarily limited to a short time frame, performed at coarse
horizontal resolutions, and/or limited in the number of sensitivity runs that can be performed (e.g., Fiore
et al., 2006;Holmes, 2018;Voulgarakis et al., 2015).
There are several alternatives (i.e., surrogate models) to running a full chemical mechanism that
capture some of the relationship between OH and trace gases, such as CO and $CH_4$, and are less
computationally expensive.  Prescribed OH fields, either static or annually-varying, from a full chemistry
simulation or a climatology have been used for decades to simulate and understand trends in CO and
$CH_4$ in a computationally-efficient way (e.g., Saito et al., 2013;Wang et al., 2004).  However, this method
linearizes CO and $CH_4$ chemistry with OH, preventing the simulation of nonlinear feedbacks in changes in
CO and $CH_4$ on OH, and thus could bias, for instance, interannual $CH_4$ changes (Chen and Prinn, 2006).
For over thirty years, parameterizations of OH have provided a viable alternative to climatologies in
helping to understand OH/CO/$CH_4$ feedbacks. Spivakovsky et al. (1990) developed a parameterization of
OH, later updated by Duncan et al. (2000), that captures many of the nonlinear feedbacks between OH
and tropospheric constituents (e.g., CO, $CH_4$, VOCs) that have loss by reaction with OH as their primary
sinks. The method to generate the parameterization uses higher order polynomials with various
chemical species, meteorological variables, and variables related to solar irradiance as inputs.  The
degree of the nonlinear impacts simulated by the parameterization of OH depends on the method used
to populate these inputs. For instance, many of the meteorological and solar irradiance variables may be
provided by the model at run time. The chemical variables that are not all simulated explicitly in the
surrogate model may be provided as climatological or monthly means from a full chemistry simulation.
Duncan et al. (2007a) and Duncan and Logan (2008) used this parameterization of OH in an atmospheric
model of CO to elucidate the causes of trends and interannual variations in CO from 1988-1997 on
regional to global scales as well as those observed by individual *in situ* monitors around the world.
Building on the CO-OH studies of Duncan et al. (2007a) and Duncan and Logan (2008), Elshorbany et
al. (2016) developed the computationally Efficient CH4–CO–OH (ECCOH) chemistry module, which
captures many of the nonlinearities and feedbacks of the $CH_4$-CO-OH system without the computational
expense of a full chemistry simulation.  ECCOH calculates 24-hour averaged OH from a combination of
archived (e.g., multiple VOCs, $NO_X$) and online (e.g., pressure, temperature, cloud albedo) chemical,
meteorological, and solar irradiance variables.  Despite the partial reliance of the parameterization of
OH in ECCOH on archived fields, its strength lies in the ability to calculate OH at a significantly reduced
computational expense (Duncan et al., 2000;Elshorbany et al., 2016).  ECCOH has been successfully
implemented in the NASA Goddard Earth Observing System (GEOS) general circulation model (GCM).

Through manipulation of the input parameters (i.e., chemical, meteorological, and solar irradiance
variables) to the parameterization of OH, as well as emissions and dynamics, ECCOH can produce
multiple, computationally cheap, sensitivity simulations that help deconvolve the causes of local to
global trends and variations in OH, CO, and $CH_4$.  For example, Strode et al. (2015) used the ECCOH
module to investigate the effects of different model biases in GEOS on simulated OH.  To do this, they
performed multiple sensitivity simulations, adjusting tropospheric water vapor, ozone, and $NO_x$ to
match satellite observations, to understand the impacts on OH and $CH_4$ lifetime.  Similarly, Elshorbany et
al. (2016) investigated the impacts of varying $CH_4$ and CO emissions on the growth rate of atmospheric
methane concentrations through multiple sensitivity runs.  One limitation of ECCOH in the configuration
used in Strode et al. (2015) and Elshorbany et al. (2016), however, is the difficulty in updating the
parameterization to reflect advances in atmospheric chemistry.
Machine learning algorithms are one potential method to quickly and accurately generate a new
parameterization of OH, offering an advance over the methods used in Duncan et al. (2000) and
Spivakovsky et al. (1990).  A variety of machine learning techniques, such as neural networks (Nicely et
al., 2017;Nicely et al., 2020;Kelp et al., 2020), ridge regression (Nowack et al., 2018), random forest
regression (Keller and Evans, 2019;Sherwen et al., 2019), and gradient boosted regression trees (GBRTs)
(Ivatt and Evans, 2020;Stirnberg et al., 2020) have been successfully used in atmospheric chemistry
applications. In particular, GBRT models (Elith et al., 2008;Chen and Guestrin, 2016) use an ensemble of
decision trees to predict the value of a target based on multiple inputs and have been used to predict
surface OH using a combination of satellite observations and output from 3-dimensional models (Zhu et
al., 2022).  Decision trees are created sequentially, with each new tree minimizing a cost function based
on the results of the previous tree (Elith et al., 2008;Stirnberg et al., 2020).  Unlike some other machine
learning algorithms, such as neural networks, regression tree methods have easily interpretable metrics
that help highlight the influence of the different input variables (Yan et al., 2016).  These metrics can
help further understanding of the model behavior in ways other machine learning techniques cannot.
GBRT models are also relatively quick to generate and can capture the highly non-linear relationships
that describe tropospheric chemistry (Ivatt and Evans, 2020).
We present a new method for the efficient generation of a parameterization of OH using GBRTs and
a full chemistry simulation from a CCM, which serves as the training dataset.  We illustrate our method
by generating a parameterization of OH for the ECCOH module (Elshorbany et al., 2016), which captures
many of the nonlinearities and feedbacks of the $CH_4$-CO-OH system, as implemented into the NASA
GEOS GCM. Our methodology allows for the parameterization to be easily and rapidly regenerated in
response to changes in, for instance, the underlying model chemical mechanism (e.g., updates to the
chemical rate constants or absorption cross-sections) or model dynamics, which affect many of the
variables that influence OH (e.g., Anderson et al., 2021).  Likewise, the parameterization can be modified
to include new input variables.  This represents a significant advance over previous, much more
laborious, methodologies to generate a parameterization of OH.  Users can and should retrain the
parameterization with datasets that are appropriate for the intended application.  That is, we are not
offering a parameterization for "off the shelf" use but a methodology by which a user can easily create a
parameterization suitable for their needs.  In Section 2, we outline the methodology used to develop the
parameterization of OH, while in Section 3, we evaluate performance of the parameterization.  Finally,
in Section 4, we summarize the results and discuss implications for scientific research.
**2.0 Description of the Methodology to Generate a Parameterization of OH**
In this section, we outline the methodology to generate a parameterization of OH that may be used
in research studies as discussed above.  Specifically, we illustrate the methodology by describing the
creation of a sample parameterization of OH for the ECCOH module that predicts daily averaged OH.  In
Section 2.1, we present the creation of the training dataset, and in Section 2.2, we outline the
methodology used to create the parameterization of OH.
**2.1  Creation of the Training Dataset for a Parameterization**
We created the training dataset using output from a 40-year (1980 -2019) GEOS CCM simulation,
consistent with our intent to integrate the parameterization into the ECCOH modeling framework.  This
simulation, called MERRA2 GMI (https://acd-ext.gsfc.nasa.gov/Projects/GEOSCCM/MERRA2GMI/), was
run in replay mode (Orbe et al., 2017) with MERRA2 (Modern Era Retrospective analysis for Research
and Applications) meteorology (Gelaro et al., 2017) and the Global Modeling Initiative (GMI) chemical
mechanism (Duncan et al., 2007b;Strahan et al., 2007).  Aerosols were calculated with the Goddard
Chemistry Aerosol Radiation and Transport (GOCART) module (Chin et al., 2002;Colarco et al., 2010).
The model was run at a resolution of c180 on the cubed sphere (approximately 0.625° longitude by 0.5°
latitude) with 72 vertical layers.  In this analysis, we use only tropospheric output at daily and monthly
resolutions. The GMI chemical mechanism includes approximately 120 species and 400 reactions,
characterizing the photochemistry of the troposphere and stratosphere. Further simulation details,
including a description of the emissions, are available elsewhere (Anderson et al., 2021;Strode et al.,
2019).
We created a dataset of training targets, representing the full range of simulated OH values, for
each month.  We generate parameterizations for each month instead of one, yearly parameterization to
increase computational efficiency of the generation of the parameterization.  The spatiotemporal
variability in the abundance and emissions of OH drivers on the yearly scale would necessitate a far
larger dataset and more complicated sampling procedures to ensure representativeness of both OH and
the input variables.  As demonstrated in Section 3.0, the adopted monthly approach accurately captures
OH while limiting the size of the training dataset.
We generated the training dataset using daily averaged data.  For each day of a month, we divided
all simulated tropospheric OH concentrations from the 40-year simulation into 20 equally-sized
percentile bins (i.e., $0 - 5^{th}$ percentile, $5^{th} - 10^{th}$ percentile, etc.).  Then, we randomly selected 200,000
values from each bin, resulting in 4,000,000 training targets for each day of the month.  We also
included the maximum and minimum OH values of the entire dataset to represent the full range of
values.  We then combined training targets to form one large dataset with 120,000,000 values (for a 30-
day month), encompassing the full range of OH concentrations from each day of the month.  To limit the
size of the training dataset, we then subsampled these targets, again randomly selecting 200,000 values
from equally-sized percentile bins of OH concentration. The procedure resulted in a dataset with
4,000,000 training targets that span all days within a given month.  A schematic of this process is shown
in Figure S1.  We omitted data from 4 years (1985, 1995, 2005, 2015) from the training dataset for
model evaluation and from an additional year, 2016, for an El Niño case study discussed in Section 3.3.
We also created a training dataset for monthly-averaged output, discussed in Sect. 4.0, using an
analogous process.
Finally, for each OH target, we extracted the inputs for the regression tree parameterization from
the MERRA2 GMI simulation from the corresponding model grid box.  We list parameterization inputs in
Table 1.  The parameterizations of Spivakovsky et al. (2000), Duncan et al. (2007a) and Elshorbany et al.
(2016), along with expert knowledge of OH chemistry, informed our choice of inputs.  The relative
location of a particular OH target is indicated with the latitude and pressure variables.  As discussed in
the next section, $NO_2$ serves as a sufficient proxy for the impact of $NO_X$ and $NO_y$ on OH, so $NO_2$ is the
only reactive nitrogen species included as an input parameter.  For both ice and water cloud as well as
aerosol optical depths, we include the optical depth above and below each datapoint as separate inputs.
We use aerosol optical depth (AOD) at 550 nm, calculated from the GOCART aerosol module.  We took
all 27 inputs from the MERRA2 GMI simulation except surface UV albedo, which we took from the Ozone
Monitoring Instrument (OMI) climatology of Kleipool et al. (2008).
*Table 1: Inputs to the machine learning parameterization of OH.  UV Albedo is the value at the surface.  Cloud fraction*
*is the fraction at a given model level.  C4 & C5 alkanes are one input as they originate from a lumped variable in the*
*GMI mechanism.*

| Chemical Inputs | | Meteorological/Radiative Inputs | |
| --- | --- | --- | --- |
| $NO_2$ | Formaldehyde (HCHO) | Temperature | Stratospheric $O_3$ Column |
| CO | Hydrogen peroxide ($H_2O_2$) | Cloud Fraction | Aerosol Optical Depth above |
| $CH_4$ | Methyl hydroperoxide ($CH_3OOH$; MHP) | Latitude | Aerosol Optical Depth below |
| $O_3$ | Acetone ($CH_3COCH_3$) | UV Albedo | Water Cloud Optical Depth above |
| Isoprene ($C_5H_8$) | C4 & C5 Alkanes | Water Vapor | Water Cloud Optical Depth below |
| Propene ($C_3H_6$) | Ethane ($C_2H_6$) | Pressure | Ice Cloud Optical Depth above |
| Propane ($C_3H_8$) | | Solar Zenith Angle | Ice Cloud Optical Depth below |

While we have used the publicly-available MERRA2 GMI dataset to train the sample
parameterization described in this manuscript, the training data could come from any simulation or
combination of self-consistent simulations that has output of the variables outlined in Table 1.  These
training datasets could come from existing simulations, which would greatly reduce computational
expense, or from a training dataset tailored for the purposes of a given study.  Even though we use
daily-averaged training data for ECCOH, a user could train the parameterization with a dataset at any
temporal resolution in order to make the parameterization compatible with a specific modeling platform
or research goal.  As discussed later, the parameterization performs best when applied to
photochemical environments analogous to those on which it was trained.  Therefore, users should
carefully ensure that the training dataset reasonably encompasses the full range of photochemical
environments necessary for a given sensitivity test or experiment.  For example, as we will discuss
further in Section 4, because the MERRA2 GMI training dataset only covers 1980 – 2018, it is
inappropriate to use this for an application exploring changes in $CH_4$ from the pre-industrial period to
2100.  Instead, a new training dataset covering that time period would be required.
**2.2 Creation of the GBRT Parameterization**
While other machine learning methods could likely produce parameterizations with similar
performance as the one we describe here, we use GBRTs because of the speed in training a new
parameterization, their accuracy, and the interpretability of the parameterization itself. We refer to the
GBRT models as parameterizations to prevent confusion when referring to 3-dimensional models.
We used the XGBoost package (Chen and Guestrin, 2016) version 0.81 in Python version 3.6 to
create 12 parameterizations of OH (one for each month), training the parameterizations on the MERRA2
GMI datasets described in Sect. 2.1.  Each parameterization outputs 24-hour averaged OH.  For each
month, we used 80% of the dataset (3.2 million datapoints) for model training and the remainder for
model validation.  In addition, as outlined in-depth in Sections 2.1 and 3.0, we also evaluated the model
on 5 years of data not included in the model training.  Increasing the size of the training dataset did not
improve model performance but did increase model training time, so the training set was restricted to a
size that represented the full ranges of OH values.
To maximize parameterization performance while also balancing the potential of overfitting, we
tuned hyperparameters, including the learning rate, the maximum tree depth, and the number of trees.
We chose hyperparameter values that minimized the parameterization normalized root mean square
error (NRMSE) (Eq. 1.) of the training dataset. In Eq. 1, N is the number of samples, y is the MERRA2 GMI
OH, $\hat{y}$ is the parameterized OH, and IQR is the interquartile range of the dataset. We set the learning
rate, which controls the magnitude of change when adding a new tree, to 0.1, while we varied the
maximum tree depth and number of trees from 6 to 22 and from 10 to 150, respectively. For both
maximum tree depth and number of trees, NRMSE initially dropped significantly with increasing value,
representing sharp improvement in parameterization performance. NRMSE values eventually
plateaued, increasing parameterization runtime without noticeably improving performance. A
combination of a maximum tree depth of 18 and 100 trees balanced performance with model training
and run time.
$$NRMSE = \frac{\sqrt{\frac{1}{N}\sum_{i=1}^{N}(\hat{y}_i - y_i)^2}}{IQR} \qquad (1)$$
We also evaluated inputs into the parameterization to ensure that each did not lead to decreased
performance, finding that no single variable dominates model performance and no variable reduces
performance. We retrained the parameterization 27 times for July, removing each input successively, to
determine its impact on the NRMSE. When we applied the resultant models to the July 2005 validation
dataset, the percentage change in the NRMSE generally increased by less than 1%. The small differences
in NRMSE indicate that there are likely variables that provide duplicate information to the
parameterization. As will be discussed in Sect. 3.2, however, the relative importance of inputs varies by
month, and some variables, though not important on average, have a large influence in specific chemical
environments. Because of these factors and a desire to use a consistent set of input variables across all
months, we did not remove any inputs from the parameterization as a result of this analysis.
Finally, we omit $NO_x$ and $NO_y$ as parameterization inputs because we find that $NO_2$ is sufficient as an
input to capture the impact of reactive nitrogen on OH in the parameterization. Because of the
importance of $NO_x$ in OH production (Spivakovsky et al., 2000;Anderson et al., 2021), we tested
performance by substituting different reactive nitrogen species for $NO_2$ as inputs to the
parameterization. We trained three additional parameterizations, including ones with $NO_x$, $NO_y$ ($NO_y$ =
NO + $NO_2$ + PAN + $2N_2O_5$ + $HNO_3$ + alkyl nitrates), and the individual $NO_y$ species. Parameterization
performance did not improve noticeably with the inclusion of $NO_x$ or the individual $NO_y$ species.
Including $NO_y$ as a group actually decreased performance.
**3.0 Evaluation of the parameterization of OH for the ECCOH module**
We now evaluate the performance of the parameterization of OH for the ECCOH module created
with the machine learning methodology. In Section 3.1, we compare the OH calculated with the
parameterization to that from the MERRA2 GMI simulation, showing agreement in both local OH
concentrations as well as in global metrics, such as $CH_4$ lifetime ($\tau_{CH4}$). In Section 3.2, we investigate the
parameterization Gain feature importance and SHapley Additive Explanations (SHAP) values to
understand the relative contributions of inputs to parameterization performance and to demonstrate
that, even though there are no physics- or chemistry-based constraints, parameterization behavior
accords with our understanding of OH chemistry. We explore the ability of the parameterization to
predict OH in response to strong deviations in its drivers from the climatological mean in Section 3.3, by
examining two El Niño events.  Finally, we note that we evaluate an offline version of the
parameterization of OH and not one integrated within the ECCOH framework.  However, the
performance will be similar based on preliminary testing and similarities in implementation to the
previous parameterization, which has been extensively evaluated (Elshorbany et al., 2016) in the GEOS
GCM.
**3.1     Ability of the parameterization to reproduce modeled OH and global OH metrics**
The parameterization is able to reproduce both the spatial distribution and concentration of
daily-averaged OH, although with noticeable errors at high latitudes in the winter hemisphere, which is
unimportant as OH is seasonally low.  Figure 1a shows the fractional difference between OH calculated
with the parameterization and OH from the MERRA2 GMI simulation for July 15, 2005, a date omitted
from the training dataset.  The parameterized and MERRA2 GMI OH fields are shown in Figure S2.  The
OH in Figure 1 has been averaged over the lower free troposphere (LFT), defined as pressures between
the top of the planetary boundary layer (PBL) and 500 hPa.  Agreement is similar throughout the
troposphere, but we highlight this region because of its importance for $CH_4$ and CO loss (Spivakovsky et
al., 2000).  For July 15, there are notable regions of bias, particularly poleward of 30° S where OH is low
(Fig. S2).  While the source of this error is unknown, it could result from a tendency of regression tree
models to have larger bias for lower values (Nowack et al., 2021).  This results in a NRMSE for the entire
troposphere of 13.9% (Fig. 2a).  At higher concentrations, the correlation between the MERRA2 GMI
simulation and the parameterized OH is much tighter than at lower concentrations, although the highest
density at all concentrations is centered around the 1:1 line.  Because the CO and $CH_4$ lifetimes are much
longer than one day, the accuracy of the parameterization on monthly timescales is more relevant to
the applications of the parameterization than an individual day.

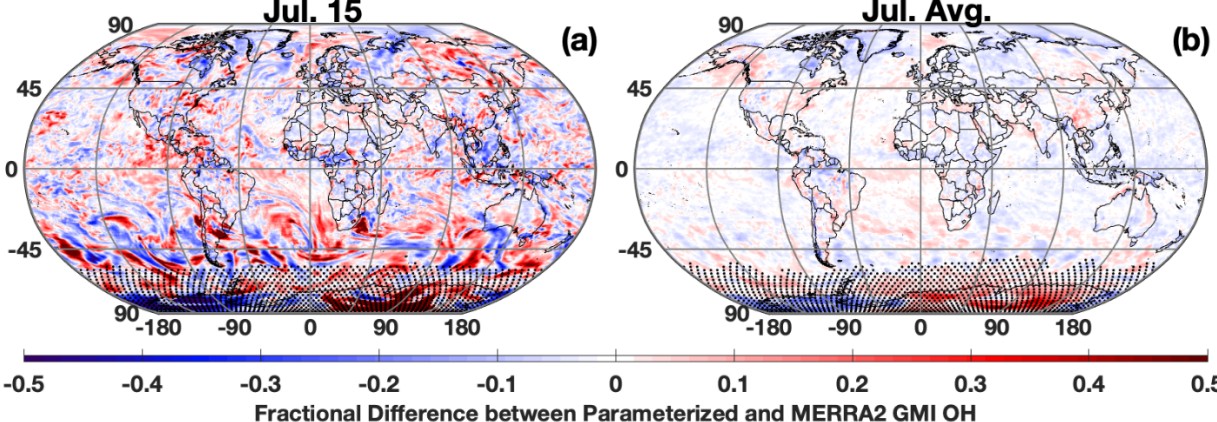

*Figure* 1*:* Fractional difference between the parameterized and MERRA2 GMI OH averaged over the LFT (top of the
PBL to 500 hPa) for July 15, 2005 (a) and averaged across all days for July 2005 (b).  Regions with low OH, defined
as a mixing ratio of less than 0.005 pptv, are indicated with stippling.
When we average the daily output to the monthly scale, the parameterization can reproduce
the global OH distribution with little error (Fig. 1-2).  For July 2005, the percentage difference between
the parameterized OH and output from the MERRA2 GMI simulation in the LFT (Fig. 1b) and throughout
the troposphere (Fig. S3) is generally within 10%, outside of the Southern Hemispheric high latitudes,
where it is polar night and OH concentrations are negligible.  The random errors evident in the daily data
in Figure 1a average out on the monthly timescale, resulting in a tight correlation ($r^2$ = 0.996) and a
NRMSE of 4.94% for all tropospheric values (Fig. 2b).  Similar results are found for the July model when
applied to other years (Table S1) and for parameterizations developed for other months (Fig. S3 and S4).
Averaging the daily output over the monthly period yields a better NRMSE by more than a factor of two
over climatology (NRMSE = 11%), defined as the mean OH from the MERRA2 GMI simulation averaged
over 1980 to 2019.

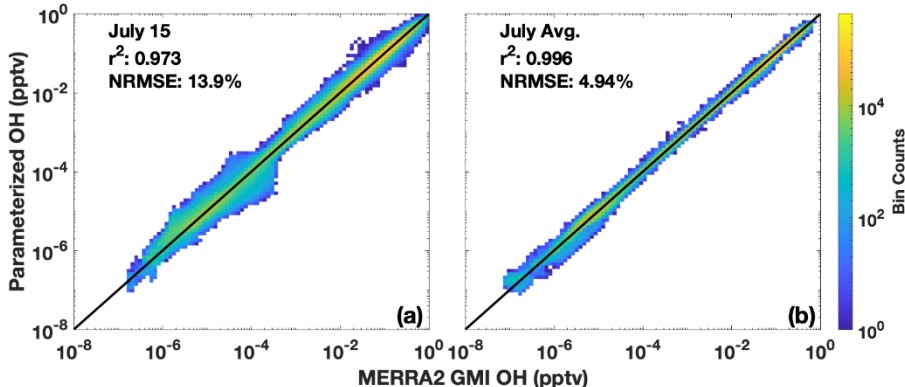

**Figure 2:** Scatter density plot of tropospheric OH from the MERRA2 GMI simulation plotted against OH calculated by
the parameterization for July 15, 2005 (a). Panel (b) shows the 24-hour averaged OH output by the parameterization
averaged across all July days for 2005. Colors indicate the number of data points in each bin. The $r^2$ of a linear least
squares regression and the NRMSE are also indicated.

In regions where global CO and $CH_4$ loss are most important, parameterization biases and errors
are low. For CO and $CH_4$, tropospheric loss to OH maximizes in the LFT in the 0 - 30° latitude band of the
summer hemisphere with near negligible loss in the winter hemisphere polar region (Fig. 3). The
comparatively large over- and underestimates over Antarctica evident in Figure 1 are irrelevant to the
OH/CO/$CH_4$ cycle because of the low loss rate in this region. In contrast, in regions where CO and $CH_4$
loss maximize, the parameterization is biased low by only -0.3 to -1.4%. The normalized absolute error
varies between 2.2% and 4.6% in the tropics and Northern Hemispheric mid-latitudes for all
tropospheric layers (MFT: pressures between 500 and 300 hPa, UFT: pressures between 300 hPa and the
tropopause). Results are similar for other months.

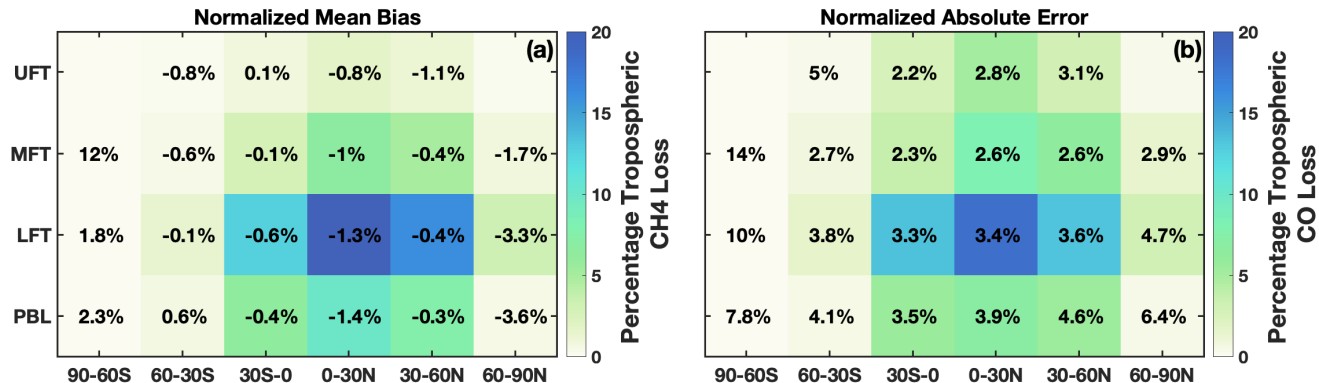

**Figure 3:** (a) Percentage of total tropospheric $CH_4$ lost to reaction with OH for July 2005 averaged over 30° zonal
mean bins and 4 atmospheric layers is shown by the background colors. The percentage loss values account for the
mass of each region relative to the total atmospheric mass. Percentages indicate the normalized mean bias of the
parameterization with respect to the MERRA2 GMI simulation. Statistics for the polar UFT are omitted because low
tropopause heights limit the data amount in these regions. (b) Same as (a) except for tropospheric CO loss and the
normalized absolute error.
The parameterization is also able to reproduce global mean metrics of OH, such as $\tau_{CH4}$, within
1.3% on average. For each month of 2005, we calculated the global, mean mass-weighted tropospheric
OH as described in Lawrence et al. (2001) and the mean tropospheric $\tau_{CH4}$ with respect to OH as
described in Nicely et al. (2020) for the MERRA2 GMI simulation, the parameterization, and the
climatological mean, defined as the average value from the MERRA2 GMI simulation between 1980 and
2019.  Results for $\tau_{CH4}$ are shown in Figure 4, and for mass-weighted OH in Figure S5.  The
parameterization captures the seasonality of the $\tau_{CH4}$, with a minimum in boreal summer and a
maximum in boreal winter.  Agreement varies slightly by month, differing by only 0.8% in January and up
to 2.5% in August, although the bias is systematically low for 2005 and the other validation years (Table
S1).  These values are reasonable and much smaller than the inter-model variability often seen in model
intercomparison projects (e.g., Nicely et al., 2020;Voulgarakis et al., 2013). Similar results are found for
the global, mean mass-weighted OH.  The Northern Hemispheric/Southern Hemispheric OH ratio (Fig.
S5) also generally agrees within 0.5% for all months, again with the exception of August.  The
comparatively weaker performance for August is limited to 2005, however, as performance of the
August parameterization in the other validation years (1985, 1995, and 2015) is closer to the 1%
difference shown by the parameterizations for the other months.  The parameterizations present a
significant improvement over the climatological mean, which for 2005, consistently underestimates $\tau_{CH4}$
for all months and by up to 6% in March.

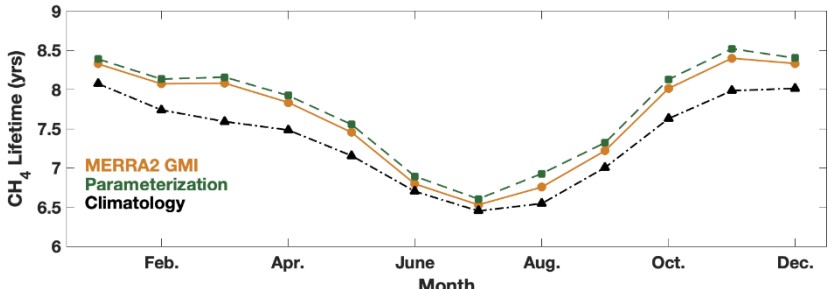

*Figure* 4*:* Global mean methane lifetime with respect to tropospheric OH from the parameterization (green squares)
and MERRA2 GMI (orange circles) for 2005 and the climatological average (black triangles) calculated from MERRA2
GMI for 1980 - 2019.
**3.2  Understanding the relative importance of input parameters**
While we have demonstrated that the parameterization is able to reproduce OH accurately, it is
also instructive to understand the relative importance the parameterization places on each of the
inputs.  Although this parameterization is not process-based, understanding how the parameterization
responds to different inputs can help determine if the regression tree is responding in a way consistent
with current understanding of OH chemistry, although there are limitations to the information that can
be gleaned from these metrics.  We evaluate the regression tree parameterization using two metrics,
the Gain feature importance as output by the XGBoost package, and SHAP values.
**3.2.1  Investigating the Gain feature importance**
The Gain feature importance (Chen and Guestrin, 2016) is a measure of the improvement in model
accuracy achieved from adding branches in the model corresponding to a specific input variable. The
Gain value therefore indicates the relative importance of each input for the model as a whole but not
for individual datapoints.  The Gain values for each input for the January and July models are shown in
Figure 5.  While there are differences between the two months, several features are similar.  Variables
that indicate geographic location (e.g., SZA, latitude, and pressure) and chemical species that have
previously shown to be dominant drivers of OH variability (e.g., $NO_2$, $O_3$, CO) and/or OH proxies (e.g.,
HCHO) (Wolfe et al., 2019;Murray et al., 2021) have some of the highest Gain values.  As we show
below, caution should be used in extrapolating results from the Gain values to a process-based
understanding of OH without prior knowledge of its response to chemical and dynamical drivers.
The relative importance of variables that indicate location is consistent with OH chemistry and
previous parameterization studies.  Primary OH production is driven by the photolysis of $O_3$ followed by
the subsequent reaction of the $O^1D$ radical, produced from that photolysis, with water vapor (e.g.,
Spivakovsky et al., 2000).  Thus, the OH distribution is strongly dependent on SZA, latitude, and
pressure.  This is consistent with the parameterization, where SZA and latitude have the highest Gain
values for both months examined here, as well as with the results of Duncan et al. (2000), who
highlighted the importance of latitude in their parameterization.
Similarly, the chemical species that are most important for controlling OH distribution on the global
scale also tend to have higher Gain values.  As discussed above, $O_3$ and $NO_X$ chemistry is instrumental in
controlling primary and secondary OH production on global scales (e.g., Spivakovsky et al.,
2000;Anderson et al., 2021), consistent with their comparatively high Gain values.  HCHO, an oxidation
product of the reaction of OH with many VOCs, has been found to be a suitable proxy for OH in the
remote atmosphere (Wolfe et al., 2019), consistent with its relative importance in both the July and
January models.

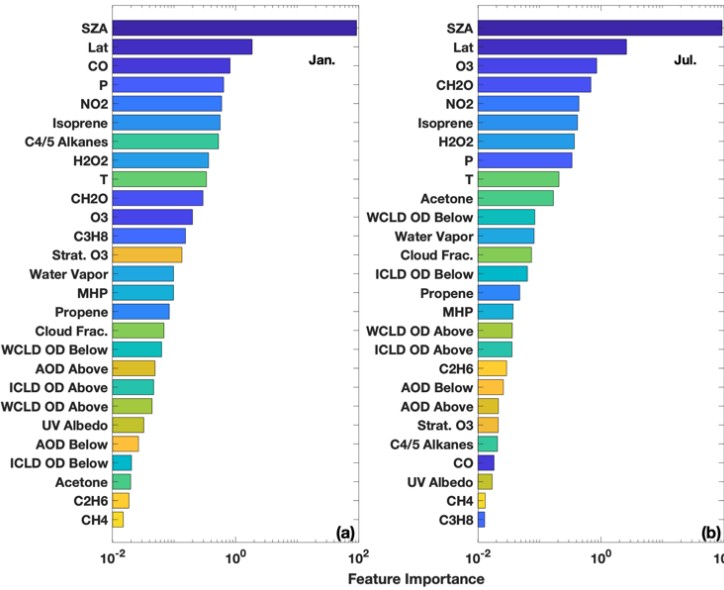

**Figure 5:** The feature importance (gains) of the January (a) and July (b) parameterizations as calculated by XGBoost.
Variables are sorted by their relative importance.  WCLD = Water cloud; ICLD = Ice Cloud; OD = Optical Depth.
"Above" and "below" for the optical depth variables indicate the optical depth above and below a particular model
grid box.  Colors are assigned to the variables to permit easier comparison of the panels.
The relative importance of global OH sinks in the parameterization, however, demonstrates the
limitations of using the Gains values to interpret the regression tree model in a process-based way.  CO,
the dominant OH sink on a global scale (Spivakovsky et al., 2000), is the most important chemical input
for the January parameterization, although it is relatively unimportant in the parameterizations for all
other months.  While tropical CO variability in MERRA2 GMI and biomass burning emissions (Duncan,
2003b) are larger in boreal winter than July, there is no process-based explanation for why CO should be
different in January from December or February. Differences in the relative importance of CO between
the two months does not imply that OH sensitivity to CO in MERRA2 GMI or the atmosphere varies in
the same manner.  Instead, the differences simply indicate that the parameterization algorithm finds CO
to be more useful in predicting OH in January than July.  Similarly, $CH_4$, the second most important OH
sink on the global scale, has low Gain values, suggesting it has little impact on model performance.  This
is likely because, in the MERRA2 GMI simulation, $CH_4$ concentrations vary little within a given latitude
band due to $CH_4$ surface concentrations being set as a boundary condition.  The methane distribution
therefore provides little additional information beyond that already contained in the variables that
indicate location.
**3.2.2. Investigating parameterization SHAP values**
While the Gain values indicate the relative importance of species in the parameterization and can
provide some information as to whether the parameterization behaves in a manner consistent with our
understanding of OH chemistry, the metric only provides information about the dataset as a whole.
Gain values can therefore obscure the importance of variables that only strongly impact the
parameterization for a small subset of the data.  To better understand the relative importance of
variables as well as the spatial variability in that importance, we also calculate the SHAP values
(Lundberg and Lee, 2017), which provide information on the relative importance of each datapoint input
into the model.
In the context of machine learning, Shapley values, an idea first developed for game theory
(Shapley, 1953), indicate the average contribution of an individual model input to all possible
combinations of inputs.  For example, to calculate the Shapley value of the variable X for a hypothetical
machine learning model with three input variables X, Y, and Z, first a model would be trained with all
three variables.  A new model would then be retrained, omitting X, and the difference between the two
models would be calculated to determine the contribution of X.  This process would then be repeated
with different permutations of input variables (e.g., X and Y, X and Z) to determine the contribution of X
in those instances.  The final Shapley value is the average of the contribution from these different
models.  SHAP values use the same concept but in a manner that reduces the computation time
(Lundberg and Lee, 2017), as this process could become prohibitive for a model, such as the
parameterization of OH, that contains 27 inputs.
We calculate SHAP values using the TreeExplainer API of the SHAP package available for Python.
One limitation of the algorithm used to calculate SHAP values is that it is too computationally expensive
to calculate the SHAP values for the tuned regression tree model.  Computational time to calculate SHAP
values for the troposphere at the native model resolution for one day is several months.  Maximizing
computational speed by degrading the model resolution and running the SHAP package with GPUs,
would take approximately 4 days for one model day.  Calculating SHAP values for a model with default
model hyperparameters, however, takes minutes.  This is due to the large reduction in the number of
trees (100 to 10) and the maximum tree depth (18 to 6) in the parameterization, which significantly
speeds up the creation of new regression trees needed in the SHAP value calculation.
We first evaluate the feasibility of using the SHAP values for the untuned model to explain the
parameterization behavior.  To test this, we created a subset of 5000 OH values from the
parameterization training dataset that spanned the full range of OH concentrations.  We then calculated
the SHAP values for the July parameterization with tuned hyperparameters as well as for a July
parameterization using the default XGBoost hyperparameters.  For the variables found to be most
important for the parameterization (e.g., SZA, $NO_2$, $O_3$, isoprene, HCHO, latitude), there are strong
correlations ($r^2$ of 0.97 or higher) for the SHAP values between the tuned and untuned model, resulting
in similar spatial distributions, although there are differences in the magnitude.  For other variables,
correlation is much weaker, although the relative importance of variables is similar for the tuned and
untuned parameterizations.  We therefore restrict our analysis primarily to variables that have highly
correlated SHAP values between the tuned and untuned models and discussion to the relative
importance of the different variables.
The distribution of SHAP values for the training dataset for July demonstrates the importance of
including each of the variables as inputs to the parameterization as well as the large variability in their
relative importance.  Figure 6 shows the distribution of the SHAP values for each input parameter of the
approximately 3.2 million datapoints used to train the July parameterization.  The median SHAP values
(Fig. 6) show similar ordering as the Gains feature importance (Fig. 5), with variables that indicate
location as well as $O_3$ and $NO_2$ being the most important in both cases.  When looking at the distribution
of the SHAP values, however, it becomes apparent that variables that appear to be unimportant for
parameterization performance in the mean (e.g., propene and $CH_4$) can have large importance for
individual datapoints.  For example, although propene can be locally important for OH chemistry, due to
its reactivity, concentrations in the remote atmosphere are low, making the species seem unimportant
in the aggregate.  Similar results are found for the January parameterization (Fig. S6).  As discussed in
Section 2.2, the SHAP values provide a rationale for including each of these species in the
parameterization.

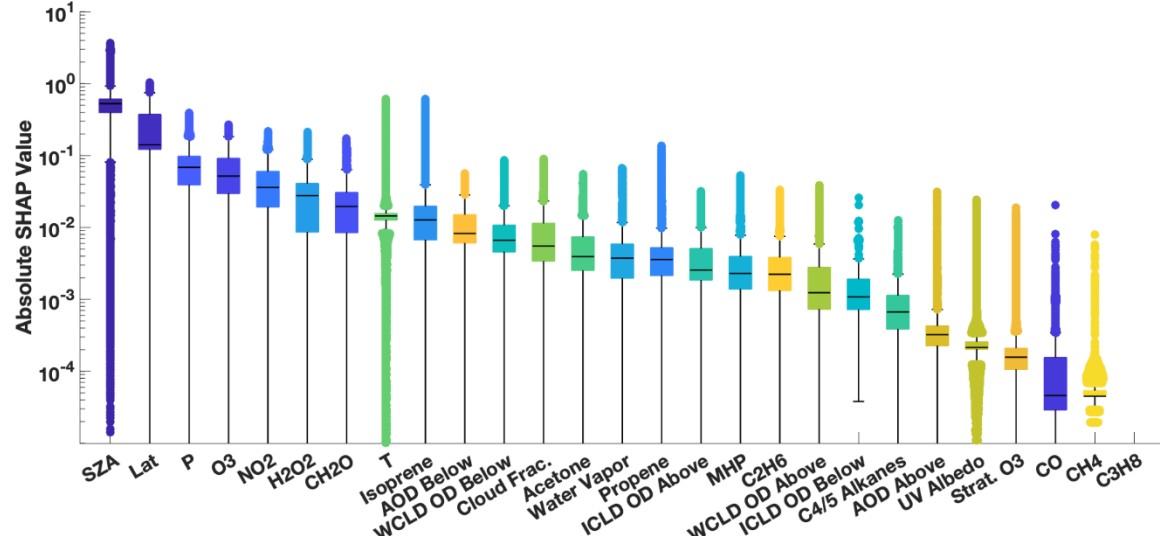

***Figure 6:*** Distribution of the absolute SHAP value for each parameterization input for July from an untuned version
of the parameterization of OH.  Input parameters are sorted by order of relative importance.  The median is indicated
with the black line, edges of the box represent the interquartile range, and whiskers represent the 5[th] and 95[th]
percentile.  Values outside this range are indicated with circles.  Note that the SHAP value for propane is zero,
indicating that it is not used by the untuned parameterization.
The SHAP values also demonstrate the spatial distribution of the relative importance of the
different chemical OH drivers.  Figure 7 shows the relative importance of $NO_2$, as determined by the
SHAP values for the untuned parameterization, for both the zonal mean and the LFT.  Note that the
untuned parameterization has large errors for low OH concentrations, so SHAP values poleward of 45 °S
should be viewed as more uncertain than those elsewhere.  In both the horizontal and vertical, the SHAP
values demonstrate that the parameterization captures the spatial pattern of the relative importance of
$NO_X$ for OH production.  The spatial pattern in Figure7a, for example, has the highest contribution of
NO₂ to the total SHAP value in the tropical UFT and in the northern hemisphere midlatitudes. This is
nearly identical to the spatial pattern of the relative contribution of the NO + HO₂ reaction to overall OH
production in the MERRA2 GMI simulation (Anderson et al., 2021). Likewise, in the LFT, the contribution
from NO₂ maximizes over continental regions with high emission and minimizes over the remote oceans.
The spatial pattern of SHAP values of isoprene also agree with OH chemistry, maximizing in regions of
strong biogenic emissions and minimizing over oceans (Fig. S7). These SHAP values demonstrate that,
although the parameterization is not process-based, its behavior at least partially accords with our
understanding of OH chemistry.

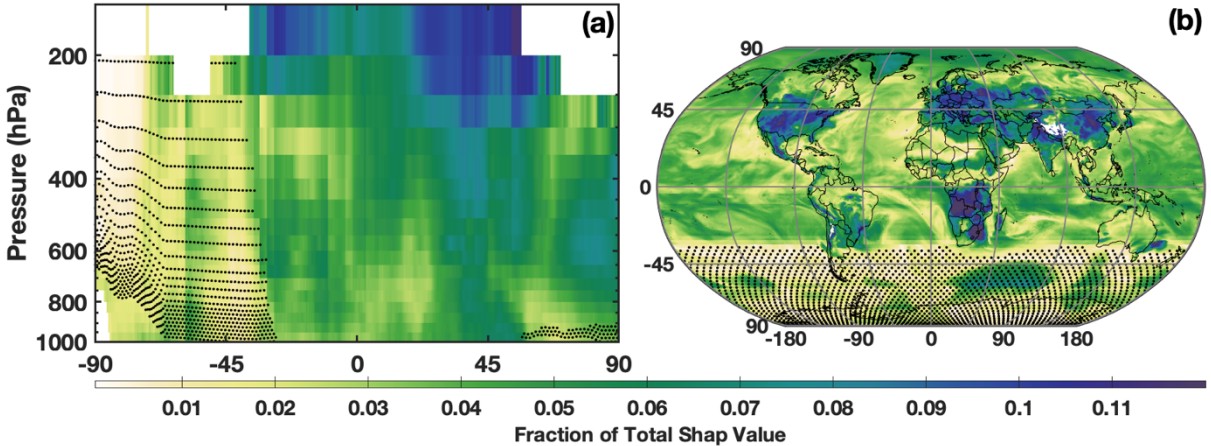

**Figure 7:** The fraction of the contribution of the NO₂ SHAP value to the sum of the absolute SHAP value of all inputs
in July is shown for the zonal mean (a) and the LFT (b). Regions where mean OH mixing ratios are below 0.03 pptv,
the point below which the untuned parameterization is unable to reasonably predict OH, are indicated by the
*stippling*.
**3.3    Case Study: Testing the parameterization response to the 2016 El Niño Event**
While we have demonstrated that the parameterization can satisfactorily reproduce OH during all
months of 2005, we now investigate its ability to capture OH accurately during the 2016 El Niño event.
which we excluded from the training dataset. Evaluating how the parameterization responds to
deviations from the climatological mean of the inputs during a large-scale event on which it was not
trained, such as the 2016 El Niño, is a strong test of its ability to predict extremes in OH as well as to
respond to deviations from the climatological mean of the parameterization inputs. The response of the
parameterized OH to these extremes in inputs will also provide a further test of the ability of the
parameterization to behave in a process-based way.
El Niño events lead to dramatic changes in the concentrations and distributions of many OH
drivers, including O₃ (Oman et al., 2011;Oman et al., 2013), CO (Duncan, 2003a;Rowlinson et al., 2019),
NOₓ (Murray et al., 2013;Murray et al., 2014) and water vapor (Shi et al., 2018;Anderson et al., 2021).
As such, the El Niño Southern Oscillation (ENSO) is the dominant mode of OH variability throughout
much of the troposphere and can result in localized changes in OH on the order of 40 – 50% from the
climatological mean (Anderson et al., 2021;Turner et al., 2018). Changes in secondary production from
NOₓ in the UFT and changes in primary production from O₃ in the PBL and LFT drive the ENSO related
variability of OH (Anderson et al., 2021). Methane emissions also vary strongly with the ENSO phase
(Zhang et al., 2018;Worden et al., 2013). In order to capture the OH/CH₄/CO system correctly, the
parameterization must be able to capture ENSO-related OH variability.

Here, we investigate the ability of the parameterization to capture OH during the El Niño events of
1997/98 and 2015/16, two of the largest such events during the period of the MERRA2 GMI simulation
according to the Multivariate ENSO Index (Wolter and Timlin, 2011). The 1997/98 event, which was
included in the training dataset, was a prototypical example of an Eastern Pacific (EP) El Niño,
characterized by sea surface temperature (SST) anomalies extending to the coast of South America. In
contrast, the 2015/16 event was a blend of an EP and a Central Pacific (CP) El Niño, also known as El
Niño Modoki, where SST anomalies extend only to the international dateline (Paek et al., 2017). These
different "flavors" of El Niño affect atmospheric distributions of OH drivers, such as water vapor (Du et
al., 2021), in different ways, suggesting different impacts on OH. While we did include other blended El
Niños (1986/87, 1987/88, and 1991/92) (Kug et al., 2009) in the training dataset, each had responses in
the atmospheric distribution of OH and its drivers distinct from the 2015/16 event. We focus our
investigation on January and the MFT, the time and location of the strongest correlation between ENSO
and OH (Anderson et al., 2021) in the MERRA2 GMI simulation. We also restrict the analysis to the OH
precursors, $NO_2$, CO, and $O_3$, as they have both a strong influence in the variability of ENSO-related OH
production/loss and have comparatively large feature importance and SHAP values in the January
parameterization.
For both the 1997/98 and 2015/16 El Niño events, each OH driver examined deviates strongly
from the climatological mean, defined as the average value from the MERRA2 GMI simulation over all
Januarys from 1980 – 2019. Both $O_3$ and $NO_2$ have pronounced positive anomalies over the western
Pacific and maritime continent and negative anomalies over the eastern Pacific (Fig. 8) that extend
throughout much of the free troposphere (Fig. S8), likely associated with changes in the Walker
Circulation as described in Oman et al. (2011). The positive anomalies over Indonesia show a distinct
westward shift during the 1997/98 event as compared to 2015/16, highlighting the variability in the
effects of ENSO on emissions and transport. CO has a large positive anomaly over much of the globe,
attributable to the increases in biomass burning during El Niño events (e.g., Duncan, 2003a). As with $O_3$
and $NO_2$, there are large differences in the spatial pattern of the CO anomalies between the two events,
particularly over the Indian Ocean, central Africa, and South America.

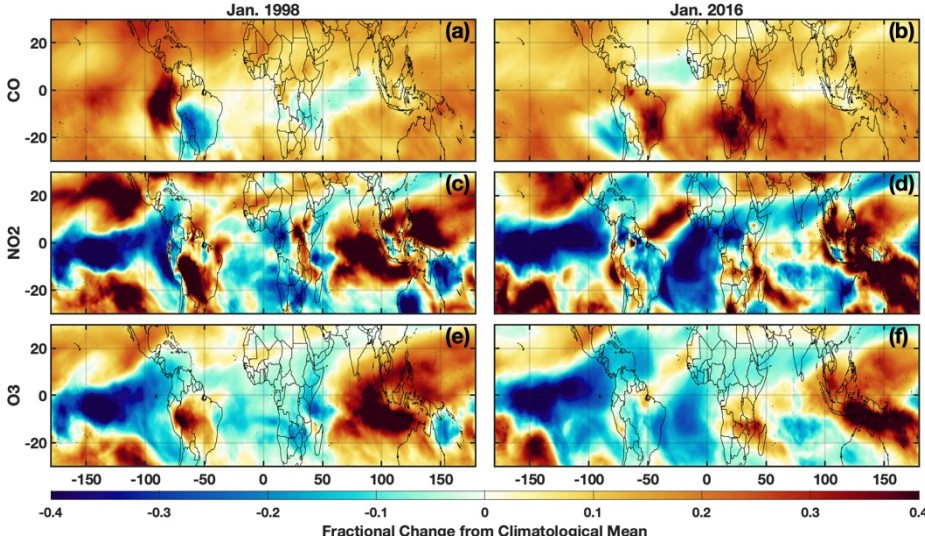

**Figure 8:** Fractional difference of the indicated variable between January 1998 (left) and the climatological mean
(1980 – 2019) calculated from the MERRA2 GMI simulation for the MFT. The same values but for January 2016 are
indicated on the right. Species shown are CO (a,b), $NO_2$ (c,d) and $O_3$ (e,f).

The differences in anomalies of the OH drivers between the 1997/98 and 2015/16 El Niño events
lead to distinct anomaly patterns in OH itself.  During the 1997/98 event, in the MFT, there are
noticeable positive anomalies in OH over much of the Indian Ocean basin, the southeastern Pacific,
South America, and the western Atlantic Ocean (Fig. 9).  During 2015/16, the positive anomalies were
more limited and most noticeable in the tropical western Pacific Ocean and southern Indian Ocean.
Along the equator, the positive anomalies extended throughout a larger portion of the troposphere
during January 1998 than 2016.  Both the parameterization inputs and the OH itself respond strongly
and in different ways to each El Niño event, providing a strong test to determine the robustness of the
parameterization.
The parameterization reproduces the ENSO-related OH anomalies for both El Niño events with
remarkable fidelity.  We ran the parameterization for all Januarys from 1980 to 2016 to calculate a
climatology and calculated the deviations for 1998 and 2016 from that value.  For both events, the
parameterization accurately captures the location and magnitude, as well as the spatial pattern, of the
OH anomalies with a few minor exceptions in the horizontal and vertical (Figs. 9 and S8).  Correlation
between the MERRA2 GMI and parameterized anomalies plotted in Figure 9 has an $r^2$ of 0.93 or higher
for both years.  The parameterization is therefore capable of reproducing both the climatological mean
of OH as well as large deviations in the mean in response to strong climatological deviations in the
model inputs, even for years excluded from the training dataset.

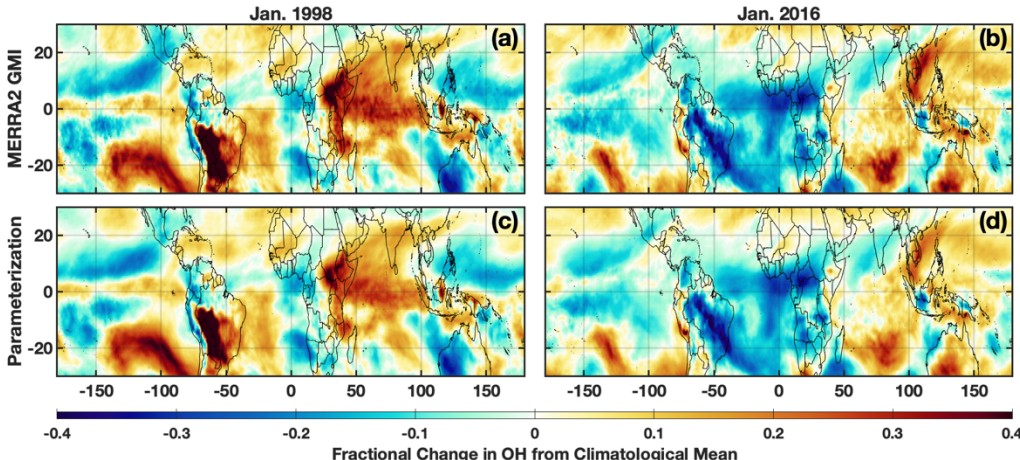

**Figure 9:** Fractional difference of the indicated variable between January 1998 (left) and the climatological mean
(1980 – 2019) calculated from the MERRA2 GMI simulation averaged over the MFT. The same values but for January
2016 are indicated on the right.  Species shown are OH from the MERRA2 GMI simulation (a,b) and OH calculated by
the parameterization (c,d).
**4.0 Discussion and Summary: The parameterization of OH as a tool for scientific research**
In this manuscript, we present a new methodology to generate a parameterization of OH that, as
compared to previous methods (e.g., Spivakovsky et al., 1990; Duncan et al., 2000), is efficient and easy
to use, allowing the user to rapidly update the parameterization of OH as necessary. The new method
uses GBRTs and a full-chemistry simulation from a CCM as the training data to generate the
parameterization of OH with a high degree of accuracy relative to the full-chemistry simulation.  We
illustrated our methodology with a parameterization designed for the ECCOH module of the GEOS CCM.
The parameterization of OH accurately reproduces OH from the full-chemistry simulation on
which it was trained, but it may not produce the desired accuracy for a given time period or scenario
outside the range represented in the training data.  Of course, the degree of degradation in accuracy
depends on how far inputs exceed the ranges of the training dataset.  In addition, the parameterization
of OH generated using inputs from one model may not be portable to another model or even a different
configuration of the same model as shown below.  The simulated relationships between OH and the
input parameters may differ because of inter-model variations in the chemical, dynamical, and radiative
schemes.  Ultimately, it is up to the user to determine an acceptable level of degradation for a specific
research topic.  In this section, we give an example of the degree of degradation in accuracy for a
parameterization of OH that uses 1) a different time period for the same model and 2) input parameters
from a different model.
**4.1: Input parameters from a different time period for the same model setup**
Analysis of a separate model simulation, the Chemistry Climate Model Initiative (CCMI) GEOS
simulation (Morgenstern et al., 2017), highlights possible limitations in extending the parameterization
to years outside of those on which it was trained, particularly if there is a strong trend in one of the
inputs.  The GEOS CCMI simulation has unconstrained meteorology, spans $1960 - 2100$, and has a
horizontal resolution of 2.0° latitude × 2.5° longitude.  Emissions for precursors of tropospheric $O_3$ and
aerosols are from the RCP6.0 scenario.  We trained two new parameterizations on the CCMI dataset,
denoted CCMI2019 and CCMI2060, using data from $1980 - 2019$ and $1980 - 2061$, respectively.  We
used the same methodology to create the training datasets as for the MERRA2 GMI parameterization.
CCMI output are only available at monthly resolution, so we trained the CCMI parameterizations on
monthly, instead of daily, averaged values.  Every 10[th] year, staring in 2000, was omitted from the
training dataset for validation.
While the CCMI2019 parameterization performed similarly to the MERRA2 GMI
parameterization for years included in the training dataset, performance degraded significantly for years
beyond 2019.  The CCMI2019 parameterization captured the $\tau_{CH4}$ for 2000 and 2010 within 1% (Fig. 10,
red line) and the NRMSE within 5% (not shown).  When we applied the parameterization to years
outside of the training window, however, performance degraded quickly and, by 2060, underestimated
$\tau_{CH4}$ by about 4%. The CCMI2060 parameterization, on the other hand, captures the $\tau_{CH4}$ lifetime within
0.5% for all validation years.
The reason for this performance degradation is likely due to input parameters that extend
beyond the range used in the training dataset.   For example, there is a strong positive trend in the
stratospheric $O_3$ column (Fig. 10), which results in chemical environments in 2060 that did not exist in
the $1980 - 2019$ training dataset.  Other variables with strong trends, such as $CH_4$ and temperature, as
well as different large-scale dynamical patterns, could also decrease parameterization performance.
These results strongly suggest caution when applying the parameterization to future scenarios outside
of the training window.  As will be discussed in the following section, care should be taken in choosing
the training dataset to ensure that it represents the full range of photochemical conditions on which the
parameterization will be applied.

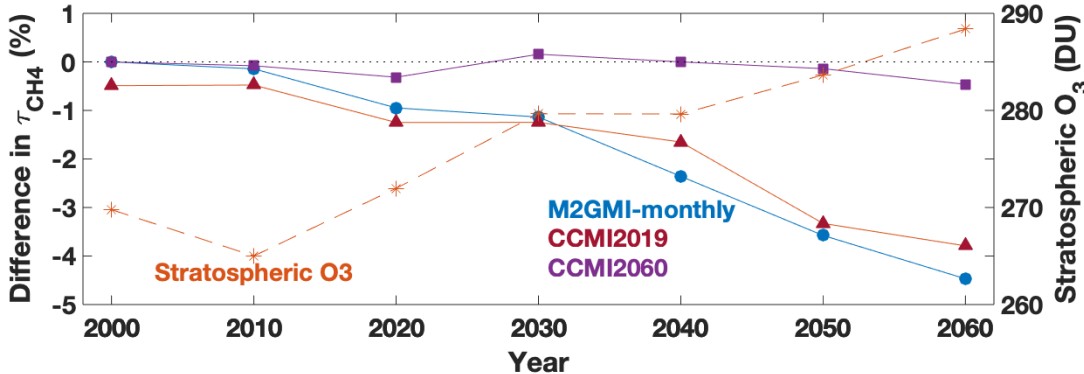

**Figure 10:** *Time series showing the percent difference between $\tau_{CH4}$ calculated from the CCMI simulation and from*
*three separate parameterizations: one trained on MERRA2 GMI output from 1980 – 2019 (blue circle), one trained*
*on CCMI output spanning 1980 – 2019 (red triangle), and one trained on CCMI output spanning 1980 – 2060 (purple*
*square). The stratospheric $O_3$ column (orange star) from the CCMI simulation averaged over 30 S to 60 N, the region*
*where tropospheric $CH_4$ loss to OH maximizes (Fig. 3), is also shown. All data are for July.*
**4.2 Input parameters from a different model setup**
Similar to applying the parameterization outside of the timeframe on which it was trained,
applying the parameterization to a different model setup also warrants caution, as the differences can
result in new chemical environments on which the parameterization was not trained. We now discuss
parameterization performance when output from the CCMI simulation discussed in Section 4.1 is input
into the MERRA2 GMI-trained parameterization. Despite both simulations being from the GEOS
framework, the CCMI simulation setup differs from the MERRA2 GMI simulation in emissions, time
frame, resolution, and meteorology (unconstrained vs specified dynamics), among others. Because
CCMI output is only available at a monthly resolution, we created a separate parameterization,
hereafter referred to as "M2GMI-monthly", using MERRA2 GMI output with identical parameterization
inputs as the daily parameterization but using monthly-averaged values. Performance is similar to that
of the parameterization trained on daily data and averaged over monthly timescales, with the NRMSE
for the troposphere on the order of 6 -7% depending on the year.
When output from the CCMI simulation is used as inputs to the M2GMI-monthly
parameterization, performance degrades significantly. For July 2000, for example, there are distinct
regions of both positive and negative biases (Fig. 11a) in parameterized OH, resulting in a NRMSE of
13%, on par with using climatology as an OH estimate. Because the largest discrepancies are centered
outside of the tropics and/or in regions with low concentrations, $\tau_{CH4}$ for year 2000 is identical between
the CCMI and parameterized OH. When applied to 2060 (Fig. 11c), which is far outside the training
period of the M2GMI-monthly parameterization, there is a near universal high bias in parameterized OH,
resulting in a NRMSE of 16% and a $\tau_{CH4}$ biased low by 4.5%. This overestimate results in a negative trend
in $\tau_{CH4}$ from parameterized OH from 2000 to 2060 (Fig. 10, blue line), despite the trend in the CCMI
simulation being positive. Applying the MERRA2 GMI parameterization to a study using the CCMI setup
would therefore misrepresent the OH/$CH_4$ cycle.

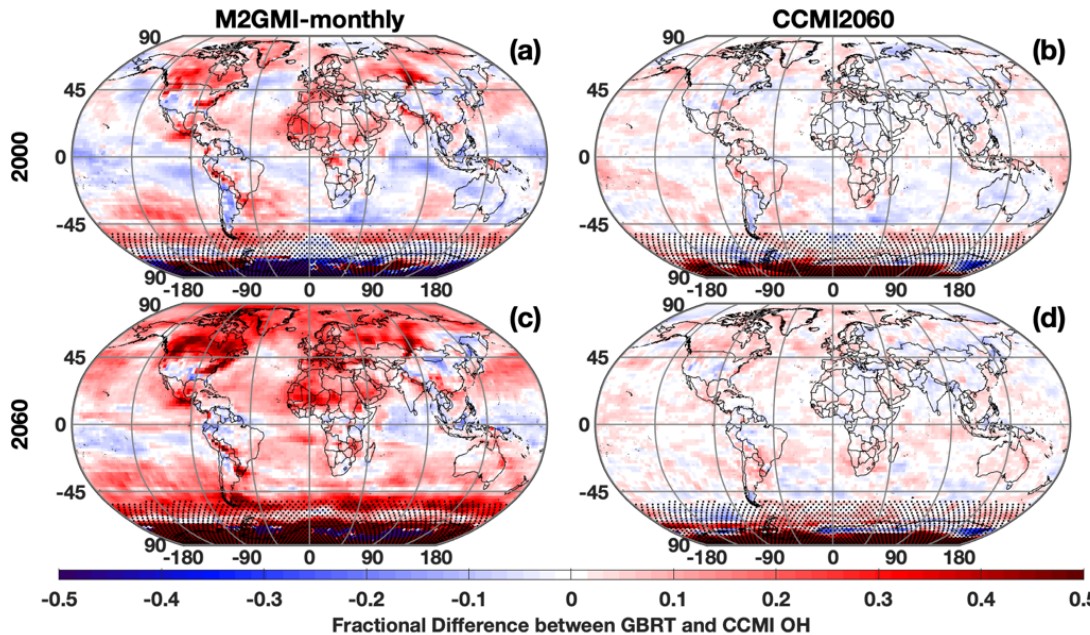

**Figure 11:** *Fractional difference between OH calculated by the M2GMI-monthly (left) and the CCMI2060 (right) parameterizations and OH output from the CCMI simulation. Data are averaged between 850 and 500 hPa for July 2000 (top) and July 2060 (bottom). Regions with low OH, defined as a mixing ratio of less than 0.005 pptv, are indicated with stippling.*

Through an analysis of the ranges of input parameters from both simulations, we found that the differences in parameterization performance for inputs from MERRA2 GMI and CCMI are likely largely attributable to differences in the stratospheric $O_3$ distributions between the two simulations. In 2060, for example, CCMI stratospheric $O_3$ has a much higher frequency of values above 300 DU than the M2GMI-monthly training dataset (Fig. 12). A smaller, but still noticeable, shift in the distribution is also found for the year 2000 (Fig. S9). Likewise, the accuracy of the M2GMI-monthly parameterization decreases from 2000 – 2060 as the stratospheric $O_3$ burden increases (Fig. 10, red line). Mechanistically, higher stratospheric ozone in CCMI should result in lower tropospheric OH because the reduction in incoming ultraviolet radiation limits tropospheric $O_3$ photolysis. This could also lead to a higher CO burden, due to the smaller OH sink. Comparisons between the OH and CO distributions from the two simulations are consistent with this hypothesis. Even though the M2GMI-monthly training dataset spanned the full range of stratospheric $O_3$ values from the CCMI simulation, the frequency of stratospheric $O_3$ values at higher concentration likely creates chemical environments in the CCMI simulation distinct from those in MERRA2 GMI, forcing the parameterization to extrapolate to a chemical space on which it was not trained. This highlights the need to compare the distribution of any parameterization inputs to that of the training dataset to ensure that the training dataset fully encompasses the range of photochemical environments necessary for a given study. Once integrated into a modeling framework, safeguards could be added to warn a user if parameterization input values fall outside of the bounds of the training dataset, as is done with the current ECCOH parameterization.

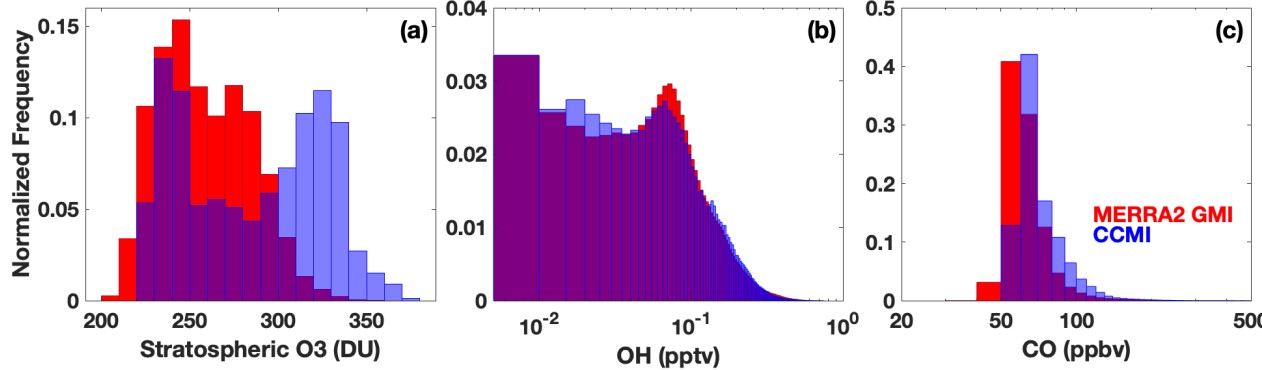

**Figure 12:** *Histograms showing the distribution of stratospheric column $O_3$ (a), tropospheric OH (b) and tropospheric CO (c) from the M2GMI-monthly parameterization training dataset (red) and from the CCMI simulation for 2060 (blue). Purple indicates areas of overlap between the two distributions.*

Again, performance improves significantly when we apply output from the CCMI simulation to the CCMI2060 parameterization. The regions of consistent high bias notable when CCMI output was applied to the M2GMI-monthly parameterization are absent for both 2000 and 2060, and NRMSE shows a factor of three improvement over the previously discussed scenario. Likewise, for all validation years, the parameterized OH resulted in a $\tau_{CH4}$ that agreed with the CCMI simulation between 0 and -0.46% (purple line, Fig. 10).

We conclude that for best performance, a separate parameterization should be created for each new modeling framework to capture OH variability accurately. This will not create an undue computational expense on an experiment. Because a full chemistry simulation is necessary to create the parameterization inputs of chemical species that are not calculated online, the necessary data to create a training dataset will already be available. The only additional time will be that required to format the regression tree inputs and to train the model, which takes approximately 2 – 3 hours for each month. This process can be performed using previously created python scripts with minimal changes. The flexibility that this modeling framework permits will facilitate its use even if there are major changes to the underlying model chemistry or dynamics.

The methodology we present here allows for the quick generation of a parameterization of OH for use in a chemistry climate model to help disentangle the complicated relationship between CO, $CH_4$, and OH. The parameterization is designed for computationally inexpensive sensitivity runs and, as such, is not designed to capture feedbacks between OH and all of its chemical and dynamical drivers (e.g. $H_2O_2$ or MHP). Instead, if a user is interested in these feedbacks, they could use the results of the sensitivity tests to identify times, locations, or chemical regimes for a targeted full chemistry simulation. Likewise, the parameterization reflects the photochemical environments of the dataset on which it was trained. Therefore, the training dataset should be carefully chosen to reflect the goals of a given study. However, we have demonstrated that the sample parameterization outlined here accurately predicts the magnitude and spatial distribution of the large deviations in OH for the 2016 El Niño, an event that was not part of the training dataset. This result gives confidence in the fidelity of a parameterization developed with our methodology to simulate the spatial and temporal responses of OH to perturbations from large variations in the chemical, dynamical, and solar irradiance drivers of OH.

**5.0 Code Availability**
The scripts used to create the training datasets and a sample script to create a parameterization have been archived by Zenodo at https://doi.org/10.5281/zenodo.6046037 (Anderson, 2022a). A sample

parameterization for the ECCOH module trained on MERRA2-GMI output is available at
https://doi.org/10.5281/zenodo.6604130 (Anderson, 2022b).

**6.0 Data Availability**

Output from the MERRA2 GMI simulation are publicly available at https://acd-ext.gsfc.nasa.gov/Projects/GEOSCCM/MERRA2GMI/.  The training dataset and training targets for the July parameterization presented here are available at https://doi.org/10.5281/zenodo.6604130 (Anderson, 2022b).  Output from the GEOSCCM simulation for CCMI is available at the Centre for Environmental Data Analysis (CED), the Natural Environment Research Council's Data Repository for Atmospheric Science and Earth Observation, at http://data.ceda.ac.uk/badc/wcrp-ccmi/data/CCMI-1/output.

**7.0 Author Contributions**

DCA wrote the manuscript, performed the data analysis, and created the parameterizations.  BND and MBFC developed the idea for the parameterization.  SAS performed three-dimensional modeling for the work.  JMN and PDI provided advice on machine learning.  All authors helped develop ideas for the analysis and contributed to the manuscript,

**8.0 Competing Interests**

The authors declare that they have no conflict of interest.

**9.0 Acknowledgements**

The authors acknowledge funding from the NASA MAP program (grant no. 80NSSC17K0220).  In addition, the authors acknowledge funding from the NASA Aura program.

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
