# Peer review of "A Machine Learning Methodology for the Generation of a Parameterization of the Hydroxyl Radical"

_Geoscientific Model Development, 2022_

## Author Comment (AC2)

We thank the reviewers for their thoughtful responses.  We have updated the manuscript to reflect their comments and address the individual comments below in red.  Updated text from the manuscript is quoted in blue.

**Reviewer 1:**
This paper describes the application of a gradient boosted regression tree machine learning approach to derive a parameterization for tropospheric OH based on CCM simulations. The approach is shown to reproduce simulated OH well under current conditions even for cases it has not been trained on, and it behaves acceptably, albeit with increasing errors, when applied to future conditions outside the standard training set. There is substantial novelty in the approach taken, and the results offer a degree of interpretability that is very interesting. The paper is generally well structured, clearly written, and appropriately illustrated. The authors have been thorough in evaluating their approach, and it is particularly good to see robust testing of input variable choice and hyperparameter value selection. The main weakness of the paper is that the potential applications of the approach are not clearly identified. What does the parameterization add beyond the simulated climatology that was used to generate it? The chemical inputs used for the parameterization are dependent on OH, and hence a full model simulation is required to capture the feedbacks. In its present form, the parameterization is not sufficient to replace CCM chemistry, and can only reproduce OH from existing simulations. If a full CCM simulation is required to generate the inputs to the parameterization, then what does the parameterization add?  If the approach is to be used as a simplified chemistry, like ECCOH, how will the oxidised inputs such as MHP and H2O2 be adjusted? A clear description of the application or purpose of the parameterization is needed, ideally with an example. Could the approach be applied with aircraft or satellite measurements to estimate OH concentrations? What is its value beyond simply reproducing existing simulations?

We have added a new discussion to the introduction (Lines 84 – 103, copied below) that more fully describes the ECCOH module for which this parameterization was designed and more explicitly highlights the applications for the parameterization and module than the previous manuscript.

> Building on the CO-OH studies of Duncan et al. (2007a) and Duncan and Logan (2008), Elshorbany et al. (2016) developed the computationally Efficient CH4–CO–OH (ECCOH) chemistry module, which captures many of the nonlinearities and feedbacks of the $CH_4$-CO-OH system without the computational expense of a full chemistry simulation.  ECCOH calculates 24-hour averaged OH from a combination of archived (e.g., multiple VOCs, $NO_x$) and online (e.g., pressure, temperature, cloud albedo) chemical, meteorological, and solar irradiance variables.  Despite the partial reliance of the parameterization of OH in ECCOH on archived fields, its strength lies in the ability to calculate OH at a significantly reduced computational expense (Duncan et al., 2000;Elshorbany et al., 2016).   ECCOH has been successfully implemented in the NASA Goddard Earth Observing System (GEOS) general circulation model (GCM).
>
> Through manipulation of the input parameters (i.e., chemical, meteorological, and solar irradiance variables) to the parameterization of OH, as well as emissions and dynamics, ECCOH can help deconvolve the causes of local to global trends and variations in OH, CO, and CH4.  For example, Strode et al. (2015) used the ECCOH module to investigate the effects of different model biases in GEOS on simulated OH.  To do this, they performed multiple sensitivity simulations, adjusting tropospheric water vapor, ozone, and $NO_x$ to match satellite observations, to understand the impacts on OH and $CH_4$ lifetime.  Similarly,

Elshorbany et al. (2016) investigated the impacts of varying $CH_4$ and CO emissions on the growth rate of atmospheric methane concentrations through multiple sensitivity runs. One limitation of ECCOH in the configuration used in Strode et al. (2015) and Elshorbany et al. (2016), however, is the difficulty in updating the parameterization to reflect advances in atmospheric chemistry.

The approach described here could indeed be used to estimate OH concentrations from *in situ* and/or satellite observations. We are actively preparing a separate manuscript on this topic to be submitted for publication in the near future. As such, we don't wish to discuss that here. We do now reference the work of Zhu et al. who use a somewhat similar approach to estimate surface OH in urban areas of the US (Lines 110 - 112):

In particular, GBRT models (Elith et al., 2008;Chen and Guestrin, 2016) use an ensemble of decision trees to predict the value of a target based on multiple inputs and have been used to predict surface OH using a combination of satellite observations and output from 3-dimensional models (Zhu et al., 2022).

Overall, the methodological approaches developed here appear sound, and the parameterization has substantial potential, but practical application of the approach is not described. The paper is not suitable for publication in GMD until the authors have made the purpose and application clear to the reader.

Specific Comments:
Title: The paper describes a machine learning methodology, but does not convincingly demonstrate a tool to improve computational efficiency, as independent application of the approach is not described. The second half of the title should be dropped.

Based on this comment and a comment from the other reviewer, we have changed the title to "A Machine Learning Methodology for the Generation of a Parameterization of the Hydroxyl Radical".

The approach is trained on an existing near present-day CCM simulation. Why was the approach not trained over a much wider range of conditions from preindustrial to possible future? A more thorough and complete set of training data would permit generation of a much more robust parameterization that was applicable to a wider range of conditions. Weaknesses in reproducing simulated future OH with the same CCM and chemistry demonstrate the frailty of the approach.

Yes, the parameterization could be trained on a much longer simulation, but our aim here is to illustrate the methodology on an existing multi-decadal simulation that we are currently using in our research. That is, we are not presenting an "off the shelf" parameterization that we wanted users to immediately integrate into their work. We have revised the text in multiple locations to make more explicit that we are offering a "recipe" to generate a parameterization, and as such, the dataset we use for training is just an example. For instance, we have added the following paragraph (Lines 197 - 210):

While we have used the publicly-available MERRA2 GMI dataset to train the sample parameterization described in this manuscript, the training data could come from any simulation or combination of self-consistent simulations that has output of the variables outlined in Table 1. These training datasets could come from existing simulations, which would greatly reduce computational expense, or from a training dataset tailored for the

purposes of a given study.  Even though we use daily-averaged training data for ECCOH, a user could train the parameterization with a dataset at any temporal resolution in order to make the parameterization compatible with a specific modeling platform or research goal.   As discussed later, the parameterization performs best when applied to photochemical environments analogous to those on which it was trained.  Therefore, users should carefully ensure that the training dataset reasonably encompasses the full range of photochemical environments necessary for a given sensitivity test or experiment.  For example, as we will discuss further in Section 4, because the MERRA2 GMI training dataset only covers 1980 – 2018, it is inappropriate to use this for an application exploring changes in $CH_4$ from the pre-industrial period to 2100.  Instead, a new training dataset covering that time period would be required.

Table 1 needs to be presented more tidily to conform to journal standards. Note also that isoprene is included twice; the first occurance should be ethane. Does UV albedo refer to UV surface albedo? Does cloud fraction apply to the column fraction above a given level or to the local fraction at that level? Optical depth above and below are separate inputs, while C4/C5 alkanes are combined; this is not clear from the Table, but is evident in Fig 5.

We have updated Table 1 (reproduced below) and the text to reflect these issues.  The caption for Table 1 now reads:

*Table 1: Inputs to the machine learning parameterization of OH.  UV Albedo is the value at the surface.  Cloud fraction is the fraction at a given model level.  C4 & C5 alkanes are one input as they originate from a lumped variable in the GMI mechanism.*

| Chemical Inputs | | Meteorological/Radiative Inputs | |
|---|---|---|---|
| $NO_2$ | Formaldehyde (HCHO) | Temperature | Stratospheric $O_3$ Column |
| CO | Hydrogen peroxide ($H_2O_2$) | Cloud Fraction | Aerosol Optical Depth above |
| $CH_4$ | Methyl hydroperoxide ($CH_3OOH$; MHP) | Latitude | Aerosol Optical Depth below |
| $O_3$ | Acetone ($CH_3COCH_3$) | UV Albedo | Water Cloud Optical Depth above |
| Isoprene ($C_5H_8$) | C4 & C5 Alkanes | Water Vapor | Water Cloud Optical Depth below |
| Propene ($C_3H_6$) | Ethane ($C_2H_6$) | Pressure | Ice Cloud Optical Depth above |
| Propane ($C_3H_8$) | | Solar Zenith Angle | Ice Cloud Optical Depth below |

In addition, we have updated the text to read (Lines 187 - 188):

For both ice and water cloud as well as aerosol optical depths, we include the optical depth above and below each datapoint as separate inputs.

Line 232: What is the likely origin of the biases poleward of 30 degrees? Could this be related to averaging of cloud cover over the diurnal cycle, a factor lost when using daily-mean input variables?

There are multiple possible sources of error.  Clouds are one, as a 24-hour averaged cloud fraction is likely not representative of the average photochemical environment.  Another possible source of error is

transport of OH chemical drivers from different latitudes and pressures. Because the parameterization strongly weights inputs that indicate location, regions that demonstrate strong vertical or horizontal mixing on a given day might also have larger errors. Unfortunately, given the nature of GBRT models, it is difficult to isolate one particular error source. Omitting individual species does not remove the bias nor does the inclusion of additional species. In addition, as the other reviewer points out, GBRTs do tend to have higher bias for smaller values. Because the primary application of the parameterization for our purposes is understanding feedbacks in the OH/CO/CH4 cycle, we feel this level of error is acceptable as it averages out on longer time scales. For an application exploring interactions between OH and shorter-lived species, a new parameterization or likely different methodology to ascertain those interactions would need to be developed.

We have updated the text as follows (Line 285 - 286):

> For July 15, there are notable regions of bias, particularly poleward of 30° S where OH is low (Fig. S2). While the source of this error is unknown, it could result from a tendency of regression tree models to have larger bias for lower values (Nowack et al., 2021).

The methane lifetime is matched well, but the small bias high is systematic throughout the year and indicates a consistent underestimate of mass- or methane-weighted OH (as also seen in Fig S5). Can the authors suggest why this arises?

This is a good point and not limited to 2005, as demonstrated in Table S1, where the parameterized OH methane lifetime is also consistently biased high. As with the larger errors poleward of 30S in Figure 1a, it is difficult to isolate the cause of the systematic bias given the "black box" nature of the machine learning model. The systematic bias is present when each individual input is removed successively from the parameterization and is also present independent of the tuning chosen for the hyperparameters. The systematic nature of the bias, however, does not invalidate the utility of the parameterization as the magnitude of the error is, as noted in the paper, generally on the order of 1%.

We have updated the text as follows (Line 338 - 340):

> Agreement varies slightly by month, differing by only 0.8% in January and up to 2.5% in August, although the bias is systematically low for 2005 and the other validation years (Table S1).

Typos and Minor Issues
Lines 74-76: Sentence grammar needs revision (or remove "Though")

We have removed "Though".

Line 174: "balance" would be clearer as "remainder" (or a similar word)

We have made this change.

Lines 35, 219 and 436: "comports" is somewhat archaic; "accords with" would be clearer

We have changed all uses of the word "comport" to "accord".

Fig 3: Red percentage labels are difficult to read over color backgrounds. Please adjust the font color so that they are legible.

We have updated this figure so that the panels are larger and that the numbers are easier to read.

Fig 5: What do the colors represent?

Colors here have no specific meaning. We colored the bars to help the reader more quickly identify the same species between figure panels and between Figures 5 & 6 since the ordering of species changes. The figure caption now reads:

> **Figure 1:** The feature importance (gains) of the January (a) and July (b) parameterizations as calculated by XGBoost. Variables are sorted by their relative importance. WCLD = Water cloud; ICLD = Ice Cloud; OD = Optical Depth. "Above" and "below" for the optical depth variables indicate the optical depth above and below a particular model grid box. Colors have no specific meaning but are specific to individual inputs for all panels of Figures 5 and 6.

Line 457: The citation for Shi et al. is missing the publication date (2018)

Thanks for pointing this out. We have made this correction.

There is a Section 4.1 (with subsections) but no Section 4.2, so renumbering of sections is needed here.

We have updated the numbering of this section.

**Reviewer 2:**
The study 'A machine learning methodology for the generation of a parameterization of the hydroxyl radical: ...' by Anderson et al. presents the results of training boosted regression trees on the output of a chemistry-climate model simulation. Specifically, the goal is to predict OH concentrations as a function of meteorological and chemical variables, which could thereafter be used as a parameterization. Mostly, this is a solid, well-written paper. In particular, most of the technical choices (following the initial choice of algorithm) are well-reasoned. However, I still have a few major and minor technical and scientific concerns, which I would like to see addressed before I would recommend publication. In particular, I am not sure how helpful such a parameterization would be if most of the chemical inputs cannot be simulated interactively. At the very least, this will help to clarify points that other readers will likely find confusing as well.
Major comments:

- My main concern relates to the motivation of building a standalone OH parameterization. I see that the authors cite several studies that used and developed OH parameterizations before and that certain inputs to those parameterizations are themselves simplified (e.g. simply time-varying climatologies). However, given the inputs for the parameterization here (e.g. NO2, ozone, VOCs) would this not pre-determine OH variations hugely, i.e. most of the expected variance in nature would not be included anyway? Is this then really just about capturing OH trends that scale with changes in CO, methane, O3? It seems to me that this is not the variance you primarily learn here, cf. your Gain values which show low contributions from e.g. CO, or CH4. If one only cares about the effect of OH on CH4,

would a simpler way not be to predict long-term CH4 changes as a function of the anyway prescribed changes in CO and O3?

As we state in our response to the first reviewer, there is likely a misunderstanding here of the applications that we have in mind for the parameterization and its intended purposes. We have clarified this by adding a new discussion of the ECCOH module and its applications (Lines 84 – 103 and reproduced above in response to the first comment of the other reviewer).

- The title is longer than necessary. Maybe consider shortening it? For example, are both 'methodology' and 'tool' needed in the same title? I am also not sure the main application of this tool would actually be in chemistry-climate model simulations. Surely, if you already run an interactive chemistry scheme it would not make a big difference to just replace OH by a parameterization? Maybe you mean something different, but it could confuse potential readers.

Based on this comment and a comment from the other reviewer, we have changed the title to "A Machine Learning Methodology for the Generation of a Parameterization of the Hydroxyl Radical".

- Training and evaluating a parameterization offline is a completely different animal from predicting 'online' ie in operational mode, especially if suddenly the inputs won't be provided by a consistent interactive chemistry model anymore. By now this is a well-known fact about machine learning parameterizations. This must be highlighted somewhere, i.e. that the study here is an offline test of the principle that requires further validation before being used operationally, especially if the OH feedbacks onto the system itself somehow (which I guess it ultimately should according to your paper title).

We agree that further evaluation of the parameterization is needed once it has been integrated into a CCM. Because we are offering a methodology to generate a parameterization, instead of a parameterization that can be directly integrated into a CCM, we felt that it was better to evaluate the parameterization itself, as the behavior once run online will likely vary depending on the CCM and the methodology used to integrate it into the CCM framework. We have added the following to the paper (Line 269 - 273):

> Finally, we note that we evaluate an offline version of the parameterization of OH and not one integrated with the ECCOH framework. However, the performance will be similar based on preliminary testing and similarities in implementation to the previous parameterization, which has been extensively evaluated (Elshorbany et al., 2016) in the GEOS GCM.

Minor comments:
- l.94-98: Please also cite Nowack et al. Using machine learning to build temperature-based ozone parameterizations for climate sensitivity simulations. Environmental Research Letters 13, 104016 (2018). https://iopscience.iop.org/article/10.1088/1748-9326/aae2be Since this was the first study to suggest machine learning parameterizations in atmospheric chemistry. Note that the authors found that ridge regression outperformed random forest regression in their case, which might have to do with ridge regression allowing for some degree of extrapolation, which you find is a potential issue here (e.g. see also Nowack et al. AMT 2021 https://amt.copernicus.org/articles/14/5637/2021/amt-14-5637-2021.pdf). Only trying out one algorithm is actually one of the main shortcomings of this study, given the aim to publish in GMD. Why not try at least a few different methods to compare their performance?

This should be computationally feasible and it is well known that there is 'no free lunch' in machine learning, so just arguing that a method is chosen because others did the same before, often in very different applications, is certainly the (too) easy way out. Another random forest application in atmospheric chemistry worth mentioning is Sherwen et al. ESSD (2019): https://essd.copernicus.org/articles/11/1239/2019/

-

We have added a reference to Nowack et al (2018) (Line 107) and Sherwen et al (2019) (Line 108).

Yes, this is an interesting question, but our main criteria in choosing a machine learning algorithm to generate the parameterization of OH were accuracy, speed, and interpretability.  We were reluctant to use linear methods, such as ridge regression, because of the highly non-linear nature of OH chemistry and felt that GBRTS more appropriately suited this particularly application.  We did perform some initial testing with neural networks, and while the accuracy of the resultant parameterization was similar to that of the GBRT parameterization, the amount of time required to train one parameterization was significantly longer for neural networks.  Because we wanted a methodology that allowed for the quick and easy regeneration of new parameterizations, we felt that neural networks would not be appropriate.  And as we indicated in the text, we wanted to use an algorithm with interpretable metrics to ensure that the model is at least roughly mimicking our process-level understanding.  As for extrapolating beyond the bounds of the training dataset, it is likely inadvisable to use any machine learning algorithm for this purpose, unless the user has thoroughly tested the resultant parameterization for the specific purposes of a given study.  This would therefore likely not result in a methodology more efficient than the one we outline here.  Finally, we note that, as described in the manuscript, the machine learning parameterization presented here can capture monthly-averaged OH within 5% with a NMB on the order of 1% for much of the troposphere, which is more than sufficient for the intended uses of the parameterization, now described in more detail in the new Introduction of the text.

We have updated the text as follows (Lines 114 - 119):

Unlike some other machine learning algorithms, such as neural networks, regression tree methods have easily interpretable metrics that help highlight the influence of the different input variables (Yan et al., 2016).  These metrics can help further understanding of the model behavior in ways other machine learning techniques cannot.  GBRT models are also relatively quick to generate and can capture the highly non-linear relationships that describe tropospheric chemistry (Ivatt and Evans, 2020).

And (Lines 213– 215):

While other machine learning methods could likely produce parameterizations with similar performance as the one we describe here, we use GBRTs because of the speed in training a new parameterization, their accuracy, and the interpretability of the parameterization itself.

- l. 101-104: yes, although the SHAP values could also be used for other methods, of course. Linear machine learning models such as ridge and Lasso are of course even more interpretable.

That is correct, although we were, as described above, reluctant to use linear techniques for such a nonlinear system.  Also, given that the SHAP value calculations were too computationally expensive for a

tuned GBRT model, it is unlikely that we could have successfully applied that method to an algorithm like a neural network. A neural network with similar performance as the GBRT parameterization required a complex architecture, which would make SHAP calculations even more computationally expensive.

- l. 106-113: my main concern here is that you train your parameterization on a single simulation and mention, correctly, that the parameterization cannot extrapolate outside the training domain. Given that we are usually interested in transient climate change scenarios, I would strongly recommend training the parameterization on a wide range of simulations (ideally at least one extremely strong forcing and one strong mitigation scenario). This would mean that re-training is expensive, unless the data already exists. Maybe that should be highlighted instead, i.e. that you could usually learn from existing simulations that will be run anyway?

We understand your concern with the training dataset. As you note below, we are offering a "recipe" for the reader to generate their own parameterization using a training dataset more appropriate for their particular use. So, the training dataset we use in this paper does not need to and cannot possibly reflect all possible scenarios under which the parameterization could be applied. As indicated elsewhere in the response to reviewers, we have updated the manuscript in multiple locations to indicate that we are presenting a methodology to generate a parameterization, not necessarily an off-the-shelf version of a parameterization. In addition, we have also added the following paragraph in Section 3.1 (Line 197 - 210):

> While we have used the publicly-available MERRA2 GMI dataset to train the sample parameterization described in this manuscript, the training data could come from any simulation or combination of self-consistent simulations that has output of the variables outlined in Table 1. These training datasets could come from existing simulations, which would greatly reduce computational expense, or from a training dataset tailored for the purposes of a given study. Even though we use daily-averaged training data for ECCOH, a user could train the parameterization with a dataset at any temporal resolution in order to make the parameterization compatible with a specific modeling platform or research goal. As discussed later, the parameterization performs best when applied to photochemical environments analogous to those on which it was trained. Therefore, users should carefully ensure that the training dataset reasonably encompasses the full range of photochemical environments necessary for a given sensitivity test or experiment. For example, as we will discuss further in Section 4, because the MERRA2 GMI training dataset only covers 1980 – 2018, it is inappropriate to use this for an application exploring changes in $CH_4$ from the pre-industrial period to 2100. Instead, a new training dataset covering that time period would be required.

- l. 122: that's a key concern: I doubt your current parameterization would work under climate change conditions, which makes it less useful. I think this point should be made very clear and highlighted prominently in the paper, ie that this is just a recipe to learn a parameterization but not a read-to-go product. Not all readers will be aware of this issue.

We have updated the manuscript in multiple locations to make it more explicitly clear that we are offering a methodology to generate a parameterization, not a ready to use version. For instance, the final paragraph of Section 1 (Line 129 - 132) now reads:

Users can and should retrain the parameterization with datasets that are appropriate for the intended application. That is, we are not offering a parameterization for "off the shelf" use but a methodology by which a user can easily create a parameterization suitable for their needs.

- l. 152: why not 2019? To avoid maximum extrapolation? I think it is good that you left out five years from the training data completely, this will reduce the risk of inflating performance measures due to spatial or temporal autocorrelation. I assume I understand your motivation correctly?

That is correct. We did not use 2019 as a validation year because we did not want to extrapolate beyond the time frame on which we trained the parameterization. Testing performed before settling on the final training time range suggests that extrapolating to 1 year outside of the training range does not affect performance, however.

- Table 1: at this point it is still unclear to me how you consider the vertical dimension, so this should be clarified earlier on (unless I overread it somehow). Do you predict OH in each tropospheric grid box (from the near-surface to the upper troposphere)? Do you include only ozone, CO etc values from the same grid box you are trying to predict, or would, for monthly-mean training data, it not be best to include at least surrounding spatial predictors? I am also not sure if, especially for surface OH, it makes sense to build one parameterization that predicts all grid points simultaneously? Wouldn't there be a spatial dependency of how e.g. ozone and OH relate, or do you think this will be sufficiently conditioned out by the other variables you include?

The parameterization calculates OH for each model grid box. The input parameters for a particular grid box are the corresponding values of the variables listed in Table 1. While we do not explicitly tell the parameterization where a particular model grid box is located, variables such as latitude and pressure generally constrain the horizontal and vertical dimensions. We have updated the text as follows to highlight these points (Line 181 - 185):

Finally, for each OH target, we extracted the inputs for the regression tree parameterization from the MERRA2 GMI simulation from the corresponding model grid box. We list parameterization inputs in Table 1. The parameterizations of Spivakovsky et al. (2000), Duncan et al. (2007a) and Elshorbany et al. (2016), along with expert knowledge of OH chemistry, informed our choice of inputs. The relative location of a particular OH target is indicated with the latitude and pressure variables.

While we have a separate parameterization for each month, the parameterization is trained on daily-averaged data, so the parameterization output is 24-hour averaged OH. As a result, we don't find it necessary to use values in neighboring grid cells as additional input parameters. While there is most definitely a spatial dependency on the relationship between OH and the input variables, both in the horizontal and vertical, the inclusion of variables indicating relative location, such as pressure, latitude, and SZA, along with variables that indicate photochemical regime (NO, O3, isoprene, various VOCs) provide a means for the machine learning model to distinguish among these different regimes.

- l. 170: why separate by month and not by location? Should not June at a SH grid location be more different from June in the NH than, say, from July at the same grid point in the SH?

Other parameterizations, including the previous one used in the ECCOH module and described in Duncan et al (2000), do break up the atmosphere into photochemical regimes. While this is one

possibility, the machine learning methodology described here does not require us to pre-determine atmospheric regimes.  Instead, the XGBoost algorithm does this implicitly by, for example, creating different branches based on isoprene concentration.  As demonstrated by Fig. 1b, this results in OH predictions with relatively small bias that is mostly randomly distributed.  It is therefore likely that breaking the atmosphere up into photochemical regimes would not significantly improve performance.  Additionally, there is substantial OH seasonal variability in OH abundance, even for a specific region, due to variability in insolation, emissions (e.g. biomass burning), etc.  A parameterization based on specific locations would still either need to be broken up into monthly parameterizations, or be trained with a dataset that encompassed the full seasonal variability of OH for that region. Because we use daily averaged data for training and the parameterization performs best when data are taken across all available days, this could result in overly large training datasets that would reduce the efficiency of generating new parameterizations, one of the goals of our methodology.

- l. 224: Random forests are known to often overpredict small values. Still wondering if this might partly be a result of throwing all data points independent of locations in one training basket?

Because the regression tree model places so much weight on the latitude and SZA variables, it is unlikely that training a separate model for these high latitude data points would change the results substantially.  We did perform a run where we omitted high latitude data from the training dataset and model performance was essentially identical to the run we present here for the rest of the globe.  We have added the following text (Line 285 – 286):

> While the source of this error is unknown, it could result from a tendency of regression tree models to have larger bias for lower values (Nowack et al., 2021).

- Figure 1: is this now tropospheric column-averaged OH? Surface OH? Please clarify. I read on: I now see that you say in the text that his is the PBL-500hPa average. Still, would add that info to the figure caption, too. Any particular reason to avoid the boundary layer? Is the spatial invariance not a good assumption there?

We have added the definition of LFT to the figure caption.  We separated the PBL from the lower free troposphere in case there were significant differences in parameterization performance due to the potentially larger impacts of emissions (e.g. biomass burning) in the PBL.  As demonstrated in Figure S3, performance in the PBL is similar to that for other levels of the atmosphere.

- At this point: do I understand correctly that you train and predict monthly-mean data? Your description in the data section was eventually somewhat confusing. So, do you train on monthly-mean data but use those functions to also predict daily OH? Might be worth highlighting again in the figure caption, for clarity.

We train on daily-averaged data, so the parameterization outputs 24-hour averaged OH.  For the monthly values, we average these daily OH values over each day of the month.  We have made this more explicitly clear in the methodology section (Line 166):

> We generated the training dataset using daily averaged data.

And also Lines 138 – 139:

> Specifically, we illustrate the methodology by describing the creation of a sample parameterization of OH for the ECCOH module that predicts daily averaged OH.

- l .250-253: implies that you tried both, ie training on daily and monthly data. Here you say that daily performs better, but above you seem to imply that daily suffers from other problems with extrapolation. What would you recommend then? Please clarify.

We have removed the reference to the model trained on monthly averaged data.  As for the accuracy of the daily model, we point out in Lines 289 - 291 that, for the purposes of understanding OH interactions with CO and CH4, the accuracy of the parameterization averaged over the monthly scale is more relevant.   So, although the NRMSE for an individual day is on the order of 14%, that error is acceptable, because averaged over a month, it drops to ~5%.

- Figure 2: clarify that this is based on daily predictions (ie based on the model trained on daily data).

The figure caption now reads:

> Scatter density plot of tropospheric OH from the MERRA2 GMI simulation plotted against OH calculated by the parameterization for July 15, 2005 (a).  Panel (b) shows the 24-hour averaged OH output by the parameterization averaged across all July days for 2005. Colors indicate the number of data points in each bin.  The $r^2$ of a linear least squares regression and the NRMSE are also indicated.

- Figure 3 should be larger, otherwise this becomes hard to read.

We have increased the size of the figure and changed the font color to make the figure more readable.

- Figure 5 and l. 351: again is this based on the model trained on daily or monthly-mean data? In any case, I would assume that CH4 has low predictability as it shows low variability on relatively short daily and monthly timescales, where other factors driving internal variability (rather than trends) are providing greater contributions to the predictions, thus showing greater feature importances. I suppose that would explain your initially maybe somewhat surprising result. If you predicted longer-term averages (e.g. decadal) I guess the picture would change dramatically. Anyway, this links to my major comment above and if much of the variability you capture will be negated if chemical inputs are derived from much simpler climatologies.

This figure, as with all the figures except those in Section 4, are based on a model trained on daily-averaged data.  As we describe above, the main application that we envision for the parameterization once integrated into the ECCOH module, is for sensitivity simulations that could, for example, help us understand the relative importance of different CO and CH4 drivers, which can easily be obtained with the current parameterization configuration.  Also, despite the low importance of CO and CH4 for the parameterization, they can still strongly affect OH in some instances, as demonstrated by the SHAP values, and the ECCOH module would still be able to capture the effects of OH on these species.

- l. 447: I see – was the motivation to exclude 2016 the specific El Nino event?

That is correct.  We have updated the text in Lines 176 - 177:

We omitted data from 4 years (1985, 1995, 2005, 2015) from the training dataset for model evaluation and from an additional year, 2016, for an El Niño case study discussed in Section 4.3.

And also in Lines 503 - 506:

Evaluating how the parameterization responds to deviations from the climatological mean of the inputs during a large-scale event on which it was not trained, such as the 2016 El Niño, is a strong test of its ability to predict extremes in OH…

- l. 542-583: I think this is a very important discussion. Random forests cannot extrapolate, so this behaviour is not surprising (e.g. Nowack et al. AMT 2021). As a more general point, I would rephrase the entire paper, also the abstract, in the sense that you present a way to learn relationships, i.e. to show that this is possible, rather than a ready-to-go parameterization, see also my other points above.

We now explicitly state in multiple locations throughout the paper that we are presenting a "recipe" to generate a parameterization and not an off-the-shelf version. We also have made clear throughout that we are evaluating a sample parameterization. References can be found at Lines 204 - 210:

As discussed later, the parameterization performs best when applied to photochemical environments analogous to those on which it was trained. Therefore, users should carefully ensure that the training dataset encompasses the full range of photochemical environments necessary for a given sensitivity test or experiment. For example, as we will discuss further in Section 4, because the MERRA2 GMI training dataset only covers 1980 – 2018, it is inappropriate to use this for an application exploring changes in $CH_4$ from the pre-industrial period to 2100. Instead, a new training dataset covering that time period would be required.

And Line 582 - 585:

The new method uses GBRTs and a full-chemistry simulation from a CCM as the training data to generate the parameterization of OH with a high degree of accuracy relative to the full-chemistry simulation. We illustrated our methodology with a parameterization designed for the ECCOH module of the GEOS CCM.

And Line 129 - 132:

Users can and should retrain the parameterization with datasets that are appropriate for the intended application. That is, we are not offering a parameterization for "off the shelf" use but a methodology by which a user can easily create a parameterization suitable for their needs.

Finally, we have added the following to the abstract (Lines 22 - 25):

To allow the user to easily target the training dataset towards a desired application, we are outlining a methodology to generate a parameterization of OH and not presenting an "off the shelf" version of a 12parameterization to be incorporated into a CCM.

- Section 4.1.2: again, not surprising because random forests cannot extrapolate. So, if you change the distribution mean, shape, and ranges, you will be in trouble. Again, selling your results as a recipe rather than a ready product, would address this point immediately.

As described above, we have updated the paper in multiple locations to indicate that we are describing a "recipe" to develop a parameterization rather than an out-of-the box version.

---

## Author Response (AR2)

We thank the reviewer for their comments. Our responses are below in red and quotations from the manuscript are in blue.

On the whole the authors have addressed the reviewers comments reasonably thoroughly and the revisions made are appropriate. Focusing on the approach (parameterization development) rather than the product (a specific parameterization) addresses many of the concerns, although this makes it less clear what the novel aspects of the study are.

My concern about the application of the approach has only partly been addressed. The use of dependent variables such as the oxidized species MHP and $H_2O_2$ as inputs prevents fully independent use of the parameterization, and means that it can only be applied reliably in chemical conditions close to those in which it was trained. This is fine for some applications (small changes, or source region tagging) but inappropriate for many others (e.g., different time periods or environments such as those illustrated in sections 4.1 and 4.2). A clearer statement of this is needed in the concluding section (see specific comments below). A key weakness of the approach is that it is not robust: there is no error checking or warning when conditions drift outside the training set, and the user can therefore never be quite sure how reliable their results are.

We have added the following text to emphasize that parameterizations of OH, such as the one described here, have been used successfully for decades. The novelty of our work is in the methodology to generate the parameterization, which is significantly less laborious than previous methods.

Line 71 - 72:

> For over thirty years, parameterizations of OH have provided a viable alternative to climatologies in helping to understand OH/CO/CH$_4$ feedbacks.

Line 107 - 109:

> Machine learning algorithms are one potential method to quickly and accurately generate a new parameterization of OH, offering an advance over the methods used in Duncan et al. (2000) and Spivakovsky et al. (1990).

Line 132 - 33:

> This represents a significant advance over previous, much more laborious, methodologies to generate a parameterization of OH.

We also note that the purpose of the parameterization, once integrated into a CCM framework, is to allow for multiple sensitivity runs, and not to capture all feedbacks on all species, such as $H_2O_2$ or MHP. We say this at Line 95 - 98:

> Through manipulation of the input parameters (i.e., chemical, meteorological, and solar irradiance variables) to the parameterization of OH, as well as emissions and dynamics, ECCOH can produce multiple, computationally cheap, sensitivity simulations that help deconvolve the causes of local to global trends and variations in OH, CO, and CH$_4$.

And in a new concluding paragraph Line 712 - 724:

The methodology we present here allows for the quick generation of a parameterization of OH for use in a chemistry climate model to help disentangle the complicated relationship between CO, CH$_4$, and OH. The parameterization is designed for computationally inexpensive sensitivity runs and, as such, is not designed to capture feedbacks between OH and all of its chemical and dynamical drivers (e.g. H$_2$O$_2$ or MHP). Instead, if a user is interested in these feedbacks, they could use the results of the sensitivity tests to identify times, locations, or chemical regimes for a targeted full chemistry simulation. Likewise, the parameterization reflects the photochemical environments of the dataset on which it was trained. Therefore, the training dataset should be carefully chosen to reflect the goals of a given study. However, we have demonstrated that the sample parameterization outlined here accurately predicts the magnitude and spatial distribution of the large deviations in OH for the 2016 El Niño, an event that was not part of the training dataset. This result gives confidence in the fidelity of a parameterization developed with our methodology to simulate the spatial and temporal responses of OH to perturbations from large variations in the chemical, dynamical, and solar irradiance drivers of OH.

Finally, in regards to there not being error checks or warnings, this is something that can and should be applied once the parameterization is integrated into the larger modeling framework, as is done with the current ECCOH parameterization. We have added the following text (Line 687 - 689):

Once integrated into a modeling framework, safeguards could be added to warn a user if parameterization input values fall outside of the bounds of the training dataset, as is done with the current ECCOH parameterization.

Fig 3: Would be better to have the same color scale on the two panels

We have updated the figure so that the limits of the colorbars are the same for both panels.

Fig 5 caption: "Colors have no specific meaning..." This is not a useful statement, and it would be better to state "Colors are assigned to the variables to permit easier comparison of the panels" (or something similar).

We have updated the figure caption with the suggested text.

Line 626-629: This sentence on application of the approach is speculative and potentially misleading. An experiment exploring the effect of a OH decrease would quickly take CH4 outside the range of the training dataset. The phrase "would require significant care to ensure valid results" should be replaced by a more honest assessment that the approach should not be used outside the training set. It would be more useful to include a statement that the training set should be chosen carefully to encompass all likely conditions under which the subsequent parameterization will be used.

We have removed the sentence and replaced it with the following (Line 630 - 632):

As will be discussed in the following section, care should be taken in choosing the training dataset to ensure that it represents the full range of photochemical conditions on which the parameterization will be applied.

Line 683: As above, the bigger message here is to ensure that the training data spans all possible conditions for which the parameterization is likely to be used. The approach should come with this as a major health warning, and its value to future users relies on them fully understanding this.

In addition to emphasizing this point in the new final paragraph (reproduced above), we have added the following text (Line 685 - 687):

> This highlights the need to compare the distribution of any parameterization inputs to that of the training dataset to ensure that the training dataset fully encompasses the range of photochemical environments necessary for a given study.

Suggestions for avoiding the frailties of the approach that are highlighted in section 4.1 and 4.2 would be valuable.

One solution is using a training dataset created from multiple simulations covering a wide range of emissions and time periods, as we currently outline in the paragraph starting on Line 201 and reproduced below. We do not employ that technique in this paper, however, because it is not necessary for our planned applications of the parameterization. Another possible way to address these issues is with a warning when values go beyond the training bounds. We have added this to the text (Line 687 - 689) and have reproduced the text above. Finally, we note again that this is a tool for sensitivity simulations and not a chemical mechanism replacement, which would likely require different machine learning techniques.

Line 201 - 214:

> While we have used the publicly-available MERRA2 GMI dataset to train the sample parameterization described in this manuscript, the training data could come from any simulation or combination of self-consistent simulations that has output of the variables outlined in Table 1. These training datasets could come from existing simulations, which would greatly reduce computational expense, or from a training dataset tailored for the purposes of a given study. Even though we use daily-averaged training data for ECCOH, a user could train the parameterization with a dataset at any temporal resolution in order to make the parameterization compatible with a specific modeling platform or research goal. As discussed later, the parameterization performs best when applied to photochemical environments analogous to those on which it was trained. Therefore, users should carefully ensure that the training dataset reasonably encompasses the full range of photochemical environments necessary for a given sensitivity test or experiment. For example, as we will discuss further in Section 4, because the MERRA2 GMI training dataset only covers 1980 – 2018, it is inappropriate to use this for an application exploring changes in $CH_4$ from the pre-industrial period to 2100. Instead, a new training dataset covering that time period would be required.